# Exophilin-8 assembles secretory granules for exocytosis in the actin cortex via interaction with RIM-BP2 and myosin-VIIa

**Fushun Fan[1†], Kohichi Matsunaga[1†], Hao Wang[1], Ray Ishizaki[1], Eri Kobayashi[1], Hiroshi Kiyonari[2,3], Yoshiko Mukumoto[3], Katsuhide Okunishi[1], Tetsuro Izumi[1,4*]**

[1]Laboratory of Molecular Endocrinology and Metabolism, Department of Molecular Medicine, Institute for Molecular and Cellular Regulation, Gunma University, Maebashi, Japan; [2]Animal Resource Development Unit, RIKEN Center for Life Science Technologies, Kobe, Japan; [3]Genetic Engineering Team, RIKEN Center for Life Science Technologies, Kobe, Japan; [4]Research Program for Signal Transduction, Division of Endocrinology, Metabolism and Signal Research, Gunma University Initiative for Advanced Research, Maebashi, Japan

**\*For correspondence:** tizumi@
gunma-u.ac.jp

[†]These authors contributed
equally to this work

**Competing interests:** The
authors declare that no
competing interests exist.

**Reviewing editor:** Suzanne R
Pfeffer, Stanford University
School of Medicine, United
States

**Abstract** Exophilin-8 has been reported to play a role in anchoring secretory granules within the actin cortex, due to its direct binding activities to Rab27 on the granule membrane and to F-actin and its motor protein, myosin-Va. Here, we show that exophilin-8 accumulates granules in the cortical F-actin network not by direct interaction with myosin-Va, but by indirect interaction with a specific form of myosin-VIIa through its previously unknown binding partner, RIM-BP2. RIM-BP2 also associates with exocytic machinery, $Ca_v1.3$, RIM, and Munc13-1. Disruption of the exophilin-8–RIM-BP2–myosin-VIIa complex by ablation or knockdown of each component markedly decreases both the peripheral accumulation and exocytosis of granules. Furthermore, exophilin-8-null mouse pancreatic islets lose polarized granule localization at the $\beta$-cell periphery and exhibit impaired insulin secretion. This newly identified complex acts as a physical and functional scaffold and provides a mechanism supporting a releasable pool of granules within the F-actin network beneath the plasma membrane.

## Introduction

Cells, including professional secretory cells, possess a peripheral microfilament web beneath the plasma membrane, referred to as the actin cortex, which maintains the cell's shape and integrity (*Orci et al., 1972*; *Aunis and Bader, 1988*). Secretory granules generated at the *trans*-Golgi network (TGN) must somehow pass through the actin cortex before they fuse with the plasma membrane and, as such, the actin cortex may act as a mechanical barrier to the exocytic site. In fact, pharmacological depolymerization of F-actin has been shown to potentiate granule exocytosis. However, such a massive disruption of the cytoskeleton may obscure physiological steps in the trafficking process. For example, cortical actin and its motor proteins, such as myosin-Va, have been suggested to function as a carrier to capture and/or transport granules to the vicinity of the plasma membrane in support of exocytosis (*Wu et al., 1998*; *Lang et al., 2000*; *Rudolf et al., 2003*; *Giner et al., 2005*; *Ivarsson et al., 2005*; *Varadi et al., 2005*; *Desnos et al., 2007*; *Wollman and Meyer, 2012*). However, the molecular mechanisms by which granules link to the F-actin network and are then processed toward exocytosis is poorly understood. Thus, a fuller understanding of the role of the cortical F-actin network in exocytosis must be established.

**eLife digest** The human body contains trillions of cells with hundreds of different jobs that must cooperate with each other. Many cells communicate using hormones and other chemical messengers that they release into the blood or tissues. These messengers are stored in containers called secretory granules, which are held just under the surface of the cell by a web of fibres made from a protein called actin. When a message needs to be sent, the granules fuse with the membrane that surrounds the cell, releasing their contents into the space outside.

A protein called exophilin-8 helps granules to fuse with the membrane. This protein attaches to both the granules and actin bundles, but its precise role is not clear. Here, Fan et al. generated mutant mice that cannot make exophilin-8 to find out what happens when this protein is missing. The experiments show that the loss of exophilin-8 prevented granules from building up at the edges of cells and releasing their contents. This was accompanied by a decrease in the amount of insulin – a hormone that regulates blood sugar levels – released by cells in the pancreas. As a result, the mutant mice had higher levels of blood sugar than normal mice. Further experiments revealed that exophillin-8 associates with a group of other proteins that work together to catch the secretory granules and anchor them to the actin bundles near to the inner edge of the cell.

If secretory granules do not fuse with the membrane properly, the chemical messages they contain are not transmitted, which can lead to disease. Since the loss of exophilin-8 affected the release of insulin from the pancreas it is possible that further work could open new avenues for diabetes research. A future challenge is to examine whether exophillin-8 also plays a similar role in the fusion of secretory granules in other cells such as nerve and immune cells, which also release a number of important chemicals.

Exophilin-8 (also known as MyRIP and Slac2-c) is a candidate molecule for the anchoring of granules within the actin cortex. It exhibits affinities both to Rab27a/b on granule membrane and to F-actin and its motor proteins, myosin-Va and -VIIa (*El-Amraoui et al., 2002*; *Fukuda and Kuroda, 2002*). In fact, overexpressed exophilin-8 has been shown to redistribute granules to the actin-rich cell periphery in the pancreatic $\beta$-cell line, MIN6 (*Mizuno et al., 2011*), and in the enterochromaffin cell line, BON (*Huet et al., 2012*). Further, knockdown of exophilin-8 decreases the number of granules beneath the plasma membrane (*Mizuno et al., 2011*; *Huet et al., 2012*) and their cargo secretion (*Waselle et al., 2003*; *Ivarsson et al., 2005*; *Mizuno et al., 2011*). These findings suggest that exophilin-8 potentiates exocytosis by locating granules within the actin cortex beneath the plasma membrane.

The present study is the first to use exophilin-8-knockout mice to demonstrate its in vivo function in glucose tolerance. Exophilin-8-null pancreatic $\beta$-cells lose polarized granule location at the cell periphery and exhibit decreased insulin secretion. We further show that exophilin-8 directly interacts with RIM-BP2, a binding protein to RIM (*Wang et al., 2000*), and that this complex formation is essential for both peripheral accumulation and efficient exocytosis of granules. In contrast to the previous proposal that exophilin-8 captures granules within the F-actin network via its direct interaction with myosin-Va (*Desnos et al., 2003*; *Huet et al., 2012*), we found that exophilin-8 does so via an indirect interaction with a specific form of myosin-VIIa through RIM-BP2. RIM-BP2 also associates with the L-type voltage-dependent $Ca^{2+}$ channel (VDCC) mediating stimulus-induced $Ca^{2+}$ influx, $Ca_v1.3$, and the priming factors, RIM and Munc13-1, in $\beta$-cells, as originally observed in neurons (*Hibino et al., 2002*; *Südhof, 2013*). Thus, the exophilin-8–RIM-BP2–myosin-VIIa complex not only physically anchors granules to the actin cortex, but may also functionally assemble molecules involved in their exocytosis. These findings reveal a previously unknown molecular mechanism in the secretory processes that efficiently retain granules beneath the plasma membrane for exocytosis.

## Results

### Exophilin-8-null β-cells exhibit decreased insulin secretion and a lower number of granules in the cell periphery

All previous studies of exophilin-8 (encoded by the *Myrip* gene) have been performed at the cellular or molecular levels. In the present study, we generated exophilin-8-knockout mice to examine its in vivo function (*Figure 1—figure supplement 1A–C*). The mutant mice were viable and fertile, with no apparent abnormalities in general appearance or behavior. However, they showed slightly reduced body weight and significantly higher blood glucose levels after a glucose load, although their insulin sensitivity was not altered (*Figure 1A*). Exophilin-8 was expressed in pancreatic islets, as well as in pituitary and brain (*Figure 1—figure supplement 1D*). Further, its absence induced decreased insulin secretion in responses to glucose, potassium, or forskolin (an activator of adenylate cyclase) with glucose (*Figure 1B–D*), but did not change secretion in response to phorbol-12-myristate-13-acetate (PMA; a protein kinase C activator) with glucose (*Figure 1E*). Cortical F-actin-disrupting PMA (*Vitale et al., 1995*) might negate the function of exophilin-8 that is localized within the actin cortex (*Desnos et al., 2003*; *Waselle et al., 2003*; *Mizuno et al., 2011*).

We then compared the distribution of insulin granules between wild-type and exophilin-8-null islets. We first coimmunostained insulin as a granule marker and Na$^+$-K$^+$ ATPase as a plasma membrane marker in isolated islets. Although the antibodies were accessible to only surface β-cells, insulin granules were preferentially polarized close to the cell edges in wild-type islets, whereas they were diffusely distributed in the perinuclear cytoplasm in exophilin-8-null islets (*Figure 2A*). Electron microscopy revealed that exophilin-8-null β-cells have a significantly lower number of granules that have centers within 300 nm of the plasma membrane (*Figure 2B,C*). Notably, however, they still exhibited granules directly attached to the plasma membrane (see arrows in *Figure 2B*).

### Exophilin-8 interacts with RIM-BP2

To understand the molecular mechanisms by which exophilin-8 functions, we investigated its interacting proteins, using the tandem affinity purification approach based on the Myc-TEV-FLAG (MEF) tag, as previously described (*Ichimura et al., 2005*; *Matsunaga et al., 2009*, *2017*). Namely, we expressed exophilin-8 fused to the MEF tag at its amino terminus in MIN6 cells and recovered the bound proteins in successive purification steps. Among the protein bands specific for the MEF-exophilin-8 eluate (*Figure 3A*), those with the highest molecular mass (150 ~ 190 kDa) were identified as RIM-BP2 and myosin-VIIa by a liquid chromatography (LC)-tandem mass spectrometry (MS/MS) analysis. Because the interaction with myosin-VIIa was already known, we further investigated that with RIM-BP2. We confirmed the presence of RIM-BP2 in the immunoprecipitate of MEF-exophilin-8 in MIN6 cells (*Figure 3B*). We also identified the endogenous complex between the two proteins in another β-cell line, INS-1 823/13 (*Figure 3C*). Although RIM-BP2 was downregulated in exophilin-8-null pancreatic islets (*Figure 3D*), it was expressed in wild-type islets and in these β-cell lines at even higher levels than in brain (*Figure 3E*), where it was initially identified (*Wang et al., 2000*; *Hibino et al., 2002*).

We next determined the binding domains responsible for the interaction between exophilin-8 and RIM-BP2. RIM-BP2 has three separate SH3 domains and three contiguous fibronectin type III (FNIII) domains (*Figure 4A*). When expressed in HEK293A cells, the first SH3 domain significantly, and the third SH3 domain strongly, interacted with exophilin-8, although the second SH3 domain or the whole FNIII domains did not (*Figure 4B*). Exophilin-8 can be divided into Rab27-binding domain (RBD), myosin-binding domain (MBD), and actin-binding domain (ABD) (*Fukuda and Kuroda, 2002*). The C-terminus of exophilin-8 containing ABD bound RIM-BP2, although the N-terminus consisting of RBD and MBD did not (*Figure 4A,C*). Exophilin-8 has two SH3 domain-interacting proline-rich sequences, RXXPXXP (*Mayer, 2001*), at residues 474–480 in the MBD and at residues 798–804 in the ABD (*Figure 4A*). Consistent with the finding of the above binding experiments (*Figure 4C*), the PA mutant replacing arginine and proline residues with alanine residues in the motif of ABD, but not that of MBD, disrupted the interaction with RIM-BP2 (*Figure 4D*). We further confirmed that wild-type exophilin-8, but not the PA (ABD) mutant, interacts with endogenous RIM-BP2 in MIN6 cells (*Figure 4E*). Taken together, these findings indicate that the two proteins interact between the first and/or third SH domains in RIM-BP2 and the C-terminal RXXPXXP sequence in exophilin-8.

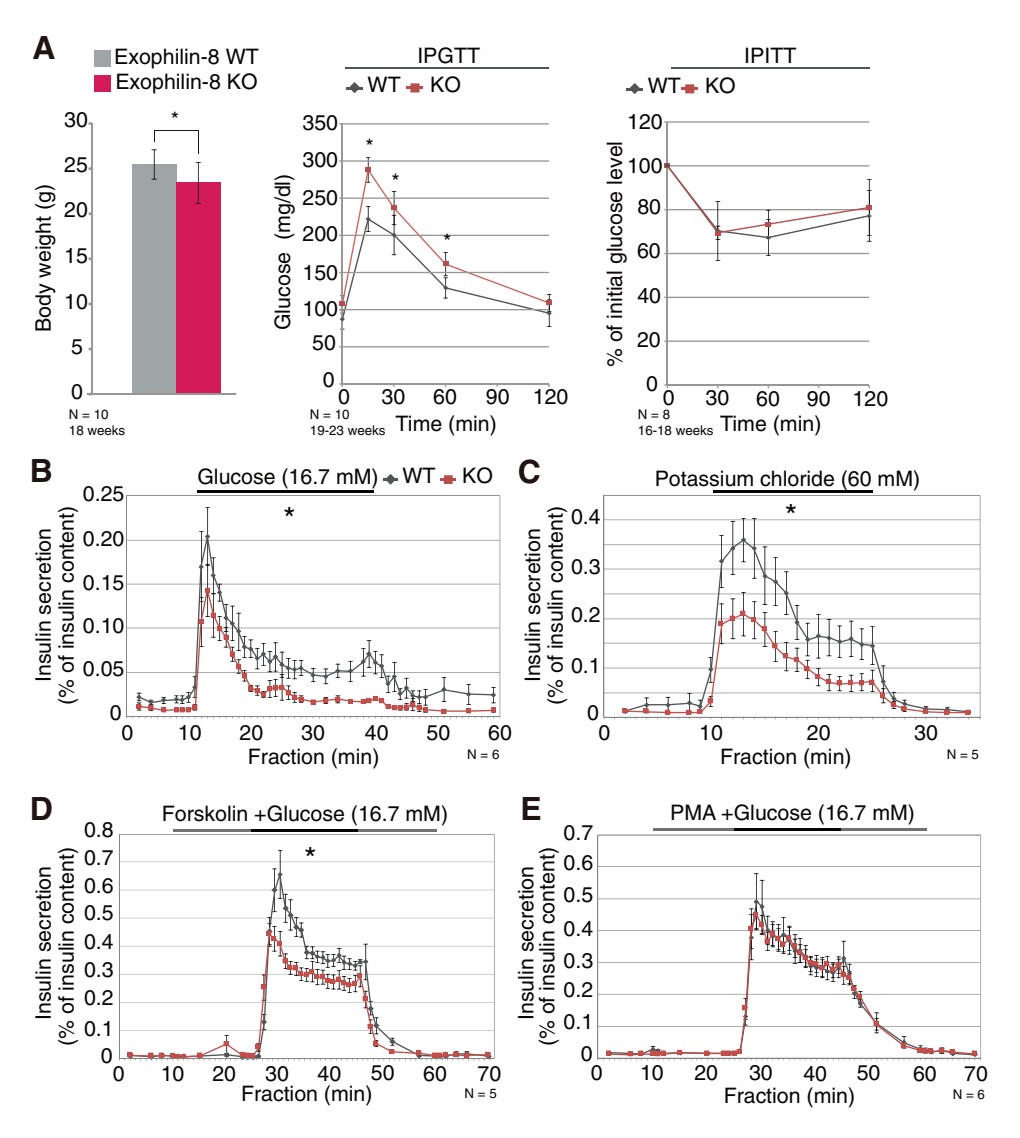

**Figure 1.** Phenotypes of exophilin-8 null mice. (**A**) In vivo phenotypes of exophilin-8-knockout (KO) mice. Each measurement was performed in age-matched, wild-type (WT; gray bars and diamonds) and KO (red bars and squares) male mice: body weight (left; 18-weeks-old, *n* = 10); blood glucose concentrations during an intraperitoneal glucose tolerance test (IPGTT; middle; 19- to 23-week-old, *n* = 10); and percentage of starting blood glucose concentration during an intraperitoneal insulin tolerance test (IPITT; right; 16- to 18-weesk-old, *n* = 8). (**B–E**) Islets isolated from age-matched (16- to 25-week-old) WT or KO male mice (*n* = 6 for B and E, *n* = 5 for C and D) were stimulated by 16.7 mM glucose for 30 min (**B**), 60 mM KCl for 15 min (**C**), or 16.7 mM glucose for 20 min (horizontal black line) in the continuous presence of either 10 μM forskolin (**D**) or 0.5 μM PMA (**E**) (horizontal black and gray lines). They were then perifused with buffer containing 2.8 mM glucose. The amount of insulin secreted into each fraction was normalized by insulin content remaining in the cells, although the latter values were not significantly different between WT and KO islets. The area under the curve during stimulation was measured. Data are means ± SEM. *p values calculated using two-tailed unpaired *t*-test are 0.031 (A, body weight), 0.0027 (A, IPGTT 0 min), $3.0 \times 10^{-7}$ (A, IPGTT 15 min), $6.6 \times 10^{-3}$ (A, IPGTT 30 min), $2.8 \times 10^{-4}$ (A, IPGTT 60 min), $7.7 \times 10^{-3}$ (B), $3.1 \times 10^{-3}$ (C), and $3.3 \times 10^{-4}$ (D), respectively.

The following figure supplement is available for figure 1:

**Figure supplement 1.** Generation of exophilin-8-null mice.

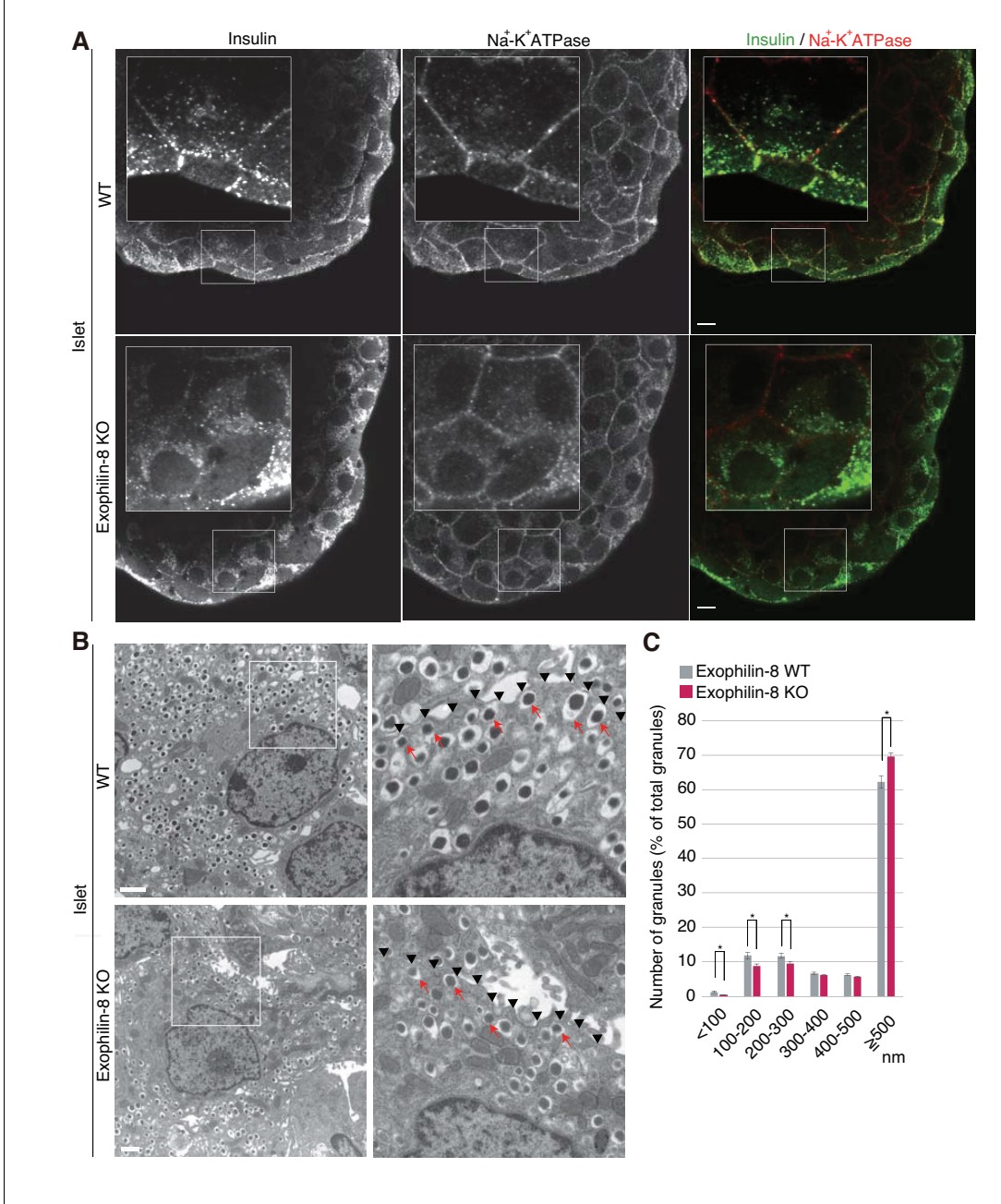

**Figure 2.** Distribution of insulin granules in exophilin-8-null $\beta$-cells. (A) Islets isolated from WT and exophilin8-KO mice were coimmunostained with anti-insulin and anti-Na$^+$-K$^+$ ATPase antibodies. Note that the antibodies were accessible to only surface $\beta$-cells. Bars, 10 µm. Insets show details at a higher magnification. (B) The isolated islets were cultured overnight and incubated in 2.8 mM glucose buffer at 37°C for 1 hr. They were then fixed and processed in a standard fashion for electron microscopy. Bar, 1 µm. Squares in left panels are shown at a higher magnification in right panels. Black arrowheads indicate the position of the plasma membrane, whereas red arrows indicate granules directly attached to the plasma membrane. (C) The distributions of insulin granules were morphometrically analyzed by electron microscopy in a total of nine $\beta$ cells from three, 20- to 22-week-old male WT (gray columns) or KO (red columns) mice (three cells from each individual mouse). All granules with centers that resided within 500 nm of the plasma membrane were categorized at 100 nm intervals. Data are shown as a percentage of the total granule number, and are shown as means ± SEM. *p values calculated using two-tailed unpaired $t$-test are 0.00182 (<100 nm), 0.01360 (100–200 nm), 0.021 (200–300 nm), and $1.4 \times 10^{-3}$ (≥500 nm), respectively.

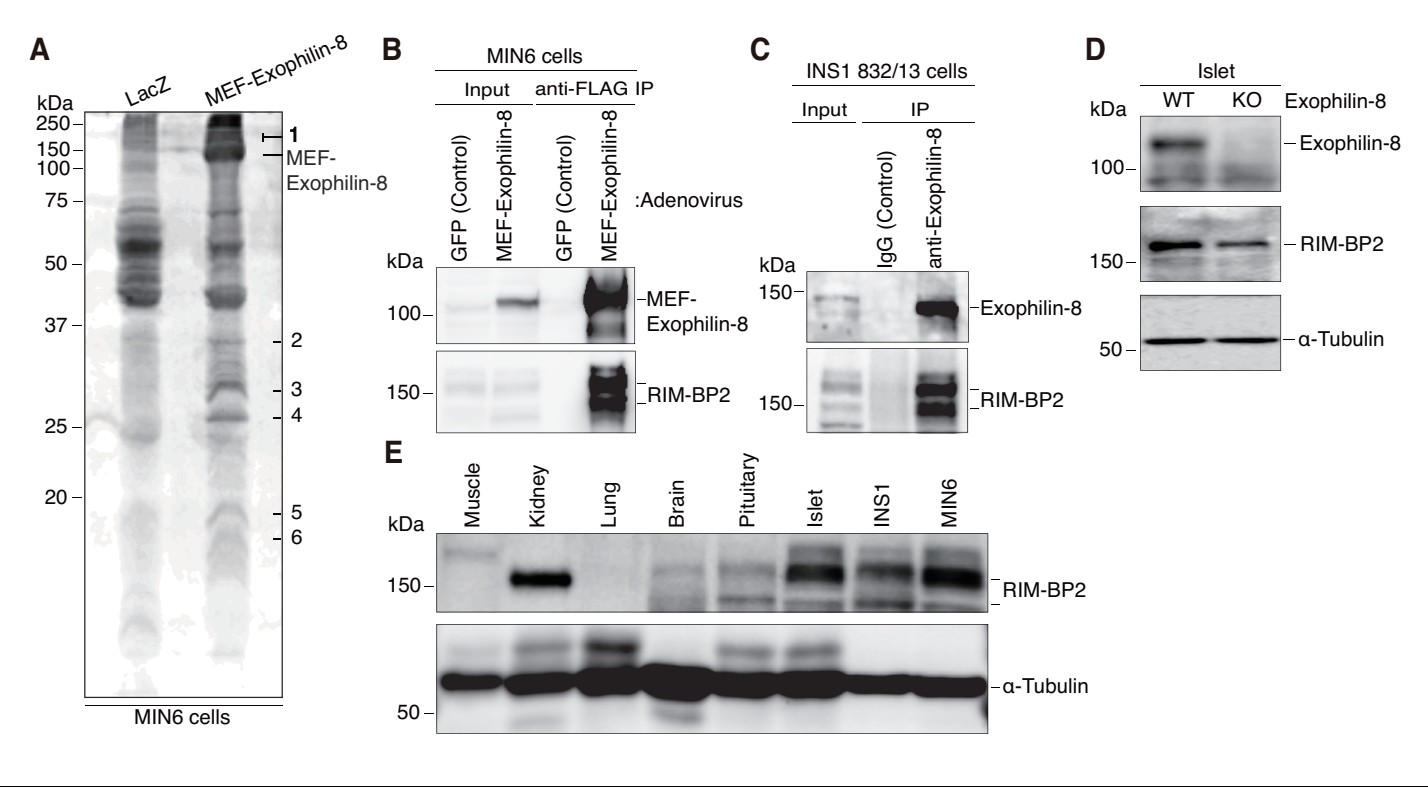

**Figure 3.** Identification of RIM-BP2 as an exophilin-8-interaction protein. (**A**) MIN6 cells expressing MEF-exophilin-8 or control LacZ protein were lysed and subjected to MEF-tag based purification. Bound proteins were detected by SDS-PAGE and Oriole fluorescent gel staining. Six bands specific to MEF-exophilin-8 are numbered. The number 1 band was identified as RIM-BP2 by LC-MS/MS analysis. (**B**) MIN6 cells were infected by adenovirus encoding MEF-exophilin-8 or control GFP protein. The immunoprecipitates (IP) with anti-FLAG antibody, as well as 1/100 of the original extracts, were immunoblotted with anti-FLAG and anti-RIM-BP2 antibodies. (**C**) The immunoprecipitates with anti-exophilin-8 antibody or control immunoglobulin G (IgG) in INS-1 832/13 cells, as well as 1/100 of the original extract, were immunoblotted with anti-exophilin-8 and anti-RIM-BP2 antibodies. (**D**) The total islet protein lysates (20 µg) from wild-type (WT) or exophilin-8-knockout (KO) mice were analyzed by immunoblotting with antibodies against exophilin-8, RIM-BP2, and α-tubulin. (**E**) The protein lysates (20 µg) from wild-type mouse tissues or cultured β-cell lines were analyzed by immunoblotting with anti-RIM-BP2 and anti-α-tubulin antibodies.

## Exophilin-8 redistributes secretory granules to the cell periphery and potentiates their exocytosis through its interaction with RIM-BP2

Exogenously expressed exophilin-8 redistributes secretory granules to the cell corners and/or the tips of cell extensions (*Mizuno et al., 2011*; *Huet et al., 2012*). Immunostaining experiments have revealed that both RIM-BP2 and exophilin-8 are endogenously colocalized with insulin granules, especially those accumulated at the cell corners, in INS-1 832/13 cells (*Figure 5A*, *Figure 5—figure supplement 1A*). Because the antibodies toward exophilin-8 and RIM-BP2 were both derived from rabbits, we investigated the colocalization of exogenously expressed exophilin-8 with endogenous RIM-BP2. We found that wild-type exophilin-8 is colocalized with RIM-BP2 and accumulates both insulin granules and RIM-BP2 at the cell corners (*Figure 5B,C*, *Figure 5—figure supplement 1B,C*). By contrast, the PA (ABD) mutant deficient in binding activity to RIM-BP2 was not polarized at the cell corners and dispersed insulin granules and RIM-BP2 diffusively in the cytoplasm. These findings indicate that exophilin-8 induces peripheral localization of insulin granules through the interaction with RIM-BP2.

To reinforce the significance of the interaction, we performed rescue experiments in islets isolated from exophilin-8-null mice. Although wild-type exophilin-8 expressed at the endogenous protein level increased glucose-stimulated insulin secretion, the PA (ABD) mutant failed to do so (*Figure 6A*, *Figure 6—figure supplement 1A,B*). Furthermore, overexpression of RIM-BP2 significantly enhanced insulin secretion in wild-type islets, but not in exophilin-8-null islets (*Figure 6B*,

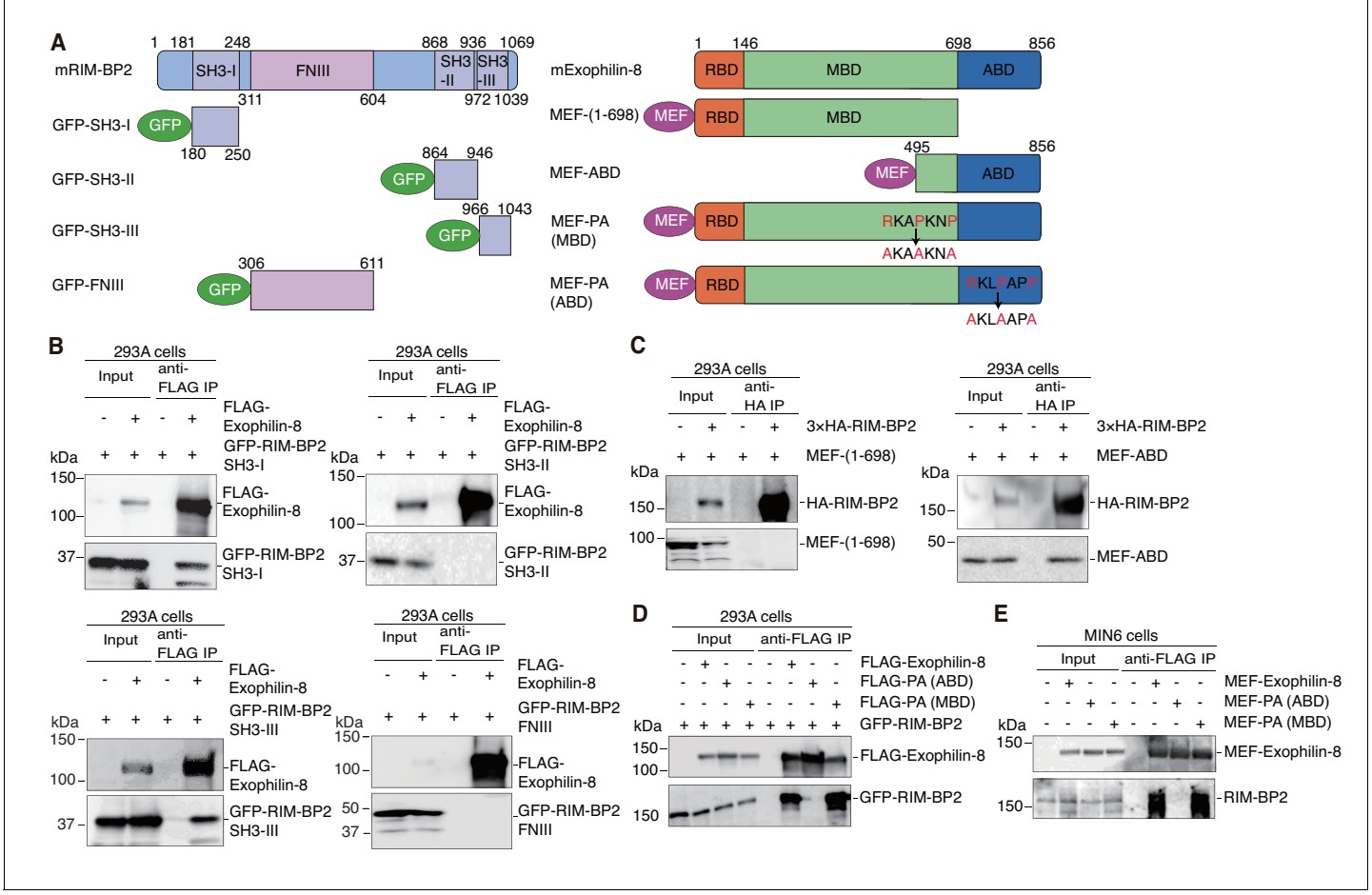

**Figure 4.** Protein domains responsible for the interaction between RIM-BP2 and exophilin-8. (A) Schematic representation of mouse RIM-BP2 (left), mouse exophilin-8 (right), and their deletion or point mutants. (B–D) HEK293A cells were transfected with plasmids encoding the indicated proteins shown in (A). The immunoprecipitates with anti-FLAG (B, D) or anti-HA (C) antibodies, as well as 1/30 of the original extracts, were immunoblotted with anti-FLAG, anti-GFP, and anti-HA antibodies. (E) MIN6 cells were infected with adenovirus encoding the indicated proteins. The immunoprecipitates with anti-FLAG antibodies, as well as 1/100 of the original extracts, were immunoblotted with anti-FLAG and anti-RIM-BP2 antibodies.

*Figure 6—figure supplement 1C*). A similar secretion-promoting effect of RIM-BP2 was observed in MIN6 cells, but not in the exophilin-8-null β-cell line (*Figure 6C*, *Figure 6—figure supplement 1D*), which was generated by a method similar to that by which the 'wild-type' MIN6 cell line was previously established (*Miyazaki et al., 1990*). We then generated RIM-BP2 lacking all the three SH3 domains (RIM-BP2ΔSH3), and confirmed that this mutant loses the binding activity to exophilin-8 (*Figure 6D*). When overexpressed in MIN6 cells, only wild-type RIM-BP2, but not RIM-BP2ΔSH3, enhanced insulin secretion (*Figure 6E*, *Figure 6—figure supplement 1E*). These findings indicate that the secretion-promoting activity of exophilin-8 or RIM-BP2 requires the interaction with the other protein.

We then examined the silencing effects of exophilin-8 or RIM-BP2 on peripheral accumulation and exocytosis of insulin granules in INS-1 832/13 cells, where robust glucose-stimulated insulin secretion occurs (*Hohmeier et al., 2000*). Although exogenous or endogenous RIM-BP2 expressed in β-cells exhibited a doublet in gels (*Figure 3E*, *Figure 6—figure supplement 1C,D*), both bands were specifically downregulated by small interfering RNA (siRNA) against RIM-BP2 (*Figure 7A*), suggesting that they represent differentially modified RIM-BP2. Exophilin-8 or RIM-BP2 knockdown profoundly decreased glucose-stimulated insulin secretion (*Figure 7B*). Furthermore, either type of knockdown induced diffusive redistributions of granules and the other protein in the cytoplasm (*Figure 7C–E*, *Figure 7—figure supplement 1*). Therefore, both exophilin-8 and RIM-BP2 are

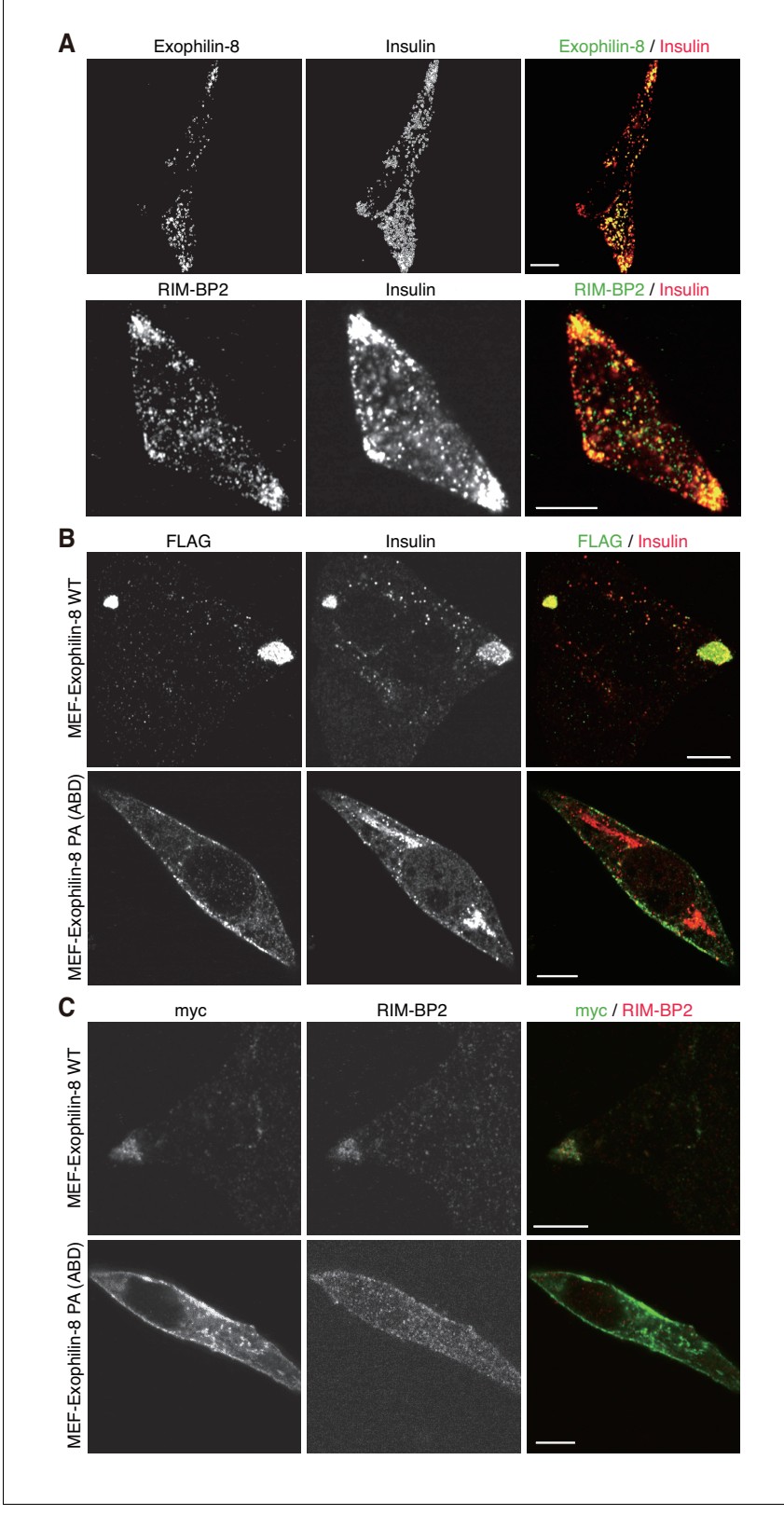

**Figure 5.** Exophilin-8 mutant deficient in binding to RIM-BP2 fails to cluster RIM-BP2 and insulin granules at cell corners. (**A**) INS-1 832/13 cells were coimmunostained with anti-insulin and either anti-exophilin-8 (upper) or anti-RIM-BP2 (lower) antibodies. (**B, C**) INS-1 832/13 cells were infected with adenovirus encoding MEF-tagged, wild-

*Figure 5 continued on next page*

*Figure 5 continued*

type (WT) or PA (ABD) mutant exophilin-8. They were coimmunostained with anti-FLAG and anti-insulin antibodies (**B**) or with anti-myc and anti-RIM-BP2 antibodies (**C**). Bars, 10 μm.

The following figure supplement is available for figure 5:

**Figure supplement 1.** More images of INS-1 832/13 cells and those expressing wild-type or PA (ABD) mutant exophilin-8.

necessary for peripheral accumulation and efficient exocytosis of secretory granules. To substantiate this conclusion, we performed rescue experiments. When wild-type or ΔSH3 RIM-BP2 was expressed in RIM-BP2-knockdown cells, only the wild type rescued the decreased insulin secretion and restored the peripheral granule accumulation (*Figure 7F,G*, *Figure 7—figure supplements 2* and *3*), again supporting the importance of the complex formation between the two proteins.

## Exophilin-8 interacts with myosin-VIIa through RIM-BP2

We then molecularly characterized the exophilin-8–RIM-BP2 complex. To reveal the exophilin-8-interacting proteins via RIM-BP2, we compared the proteins coimmunoprecipitated with MEF-tagged, wild-type exophilin-8 with those from the PA (ABD) mutant, in the presence of hemagglutinin (HA)-tagged RIM-BP2 in INS-1 832/13 cells. As shown in *Figure 8A*, the wild type, but not the PA (ABD) mutant, coprecipitated RIM-BP2, and concordantly interacted with RIM-BP2-interacting $Ca_v1.3$ and RIM, and RIM-interacting Munc13-1, as has been observed in neurons (*Wang et al., 2000*; *Betz et al., 2001*; *Hibino et al., 2002*). By contrast, both the wild type and the mutant interacted with previously known exophilin-8-interacting proteins, such as Rab27a, protein kinase A (PKA), Sec6 (*Goehring et al., 2007*), and actin. However, myosin-Va was not found in the immunoprecipitate of either wild-type or mutant exophilin-8, but was easily detected in that of exophilin-3 (also known as melanophilin and Slac2a; *Figure 8B*), another Rab27 effector that exhibits binding activity to myosin-Va (*Fukuda et al., 2002*; *Nagashima et al., 2002*; *Strom et al., 2002*; *Wu et al., 2002*). Instead, we identified myosin-VIIa in the immunoprecipitate, the tail domain of which has been reported to interact with exophilin-8 in both heterologous cells and in vitro (*El-Amraoui et al., 2002*; *Fukuda and Kuroda, 2002*). Unexpectedly, however, the PA (ABD) exophilin-8, despite the mutation outside of MBD, failed to interact with myosin-VIIa. This finding indicates that the same proline motif of exophilin-8 is involved in binding with myosin-VIIa as well as with RIM-BP2, or that exophilin-8 indirectly interacts with myosin-VIIa through RIM-BP2. To determine which was the case, we investigated whether the interaction between RIM-BP2 and myosin-VIIa occurs in the absence of exophilin-8. We found that the interaction persists in the exophilin-8-null β-cell line and is preserved by the exogenous expression of wild-type exophilin-8 (*Figure 8C*), indicating that RIM-BP2, but not exophilin-8, is primarily involved in the interaction with myosin-VIIa in cells.

To further investigate the interactions, we simultaneously expressed these proteins in HEK293A cells. The One-STrEP-FLAG (OSF)-tag was attached to wild-type or PA mutant exophilin-8 ABD, and the binding proteins were pulled down using Strept-Tactin beads. As expected, wild-type, but not PA mutant, exophilin-8 ABD bound RIM-BP2, strongly indicating the direct interaction because there were no other specific proteins (*Figure 9A*, lanes 2 and 5). Unexpectedly, however, both wild-type and PA mutant exophilin-8 ABD bound myosin-VIIa with or without RIM-BP2 expression (*Figure 9A*, lanes 3, 4, 6, and 7), in contrast to the finding in INS-1 832/13 cells (*Figure 8B*). Therefore, exophilin-8 seems to directly interact with myosin-VIIa at least in heterologous cells, consistent with the previous findings (*El-Amraoui et al., 2002*; *Fukuda and Kuroda, 2002*), and furthermore, this interaction does not require the MBD or the proline-rich motif in the ABD of exophilin-8. We then expressed OST-tagged RIM-BP2 and the binding proteins were examined. We confirmed that RIM-BP2 bound wild-type, but not PA mutant, exophilin-8 ABD (*Figure 9B*, lanes 2 and 3). However, RIM-BP2 could not bind myosin-VIIa without simultaneous expression of wild-type exophilin8 ABD (*Figure 9B*, lanes 4–6), indicating that RIM-BP2 cannot directly bind myosin-VIIa. We noticed that, although the approximate molecular mass by sodium dodecyl sulfate (SDS)-polyacrylamide gel electrophoresis (PAGE) of myosin-VIIa expressed in HEK293A cells is ~260 kDa corresponding to its calculated molecular mass (*Figure 9A,B*), that of myosin VIIa interacting with RIM-BP2 in β-cell lines

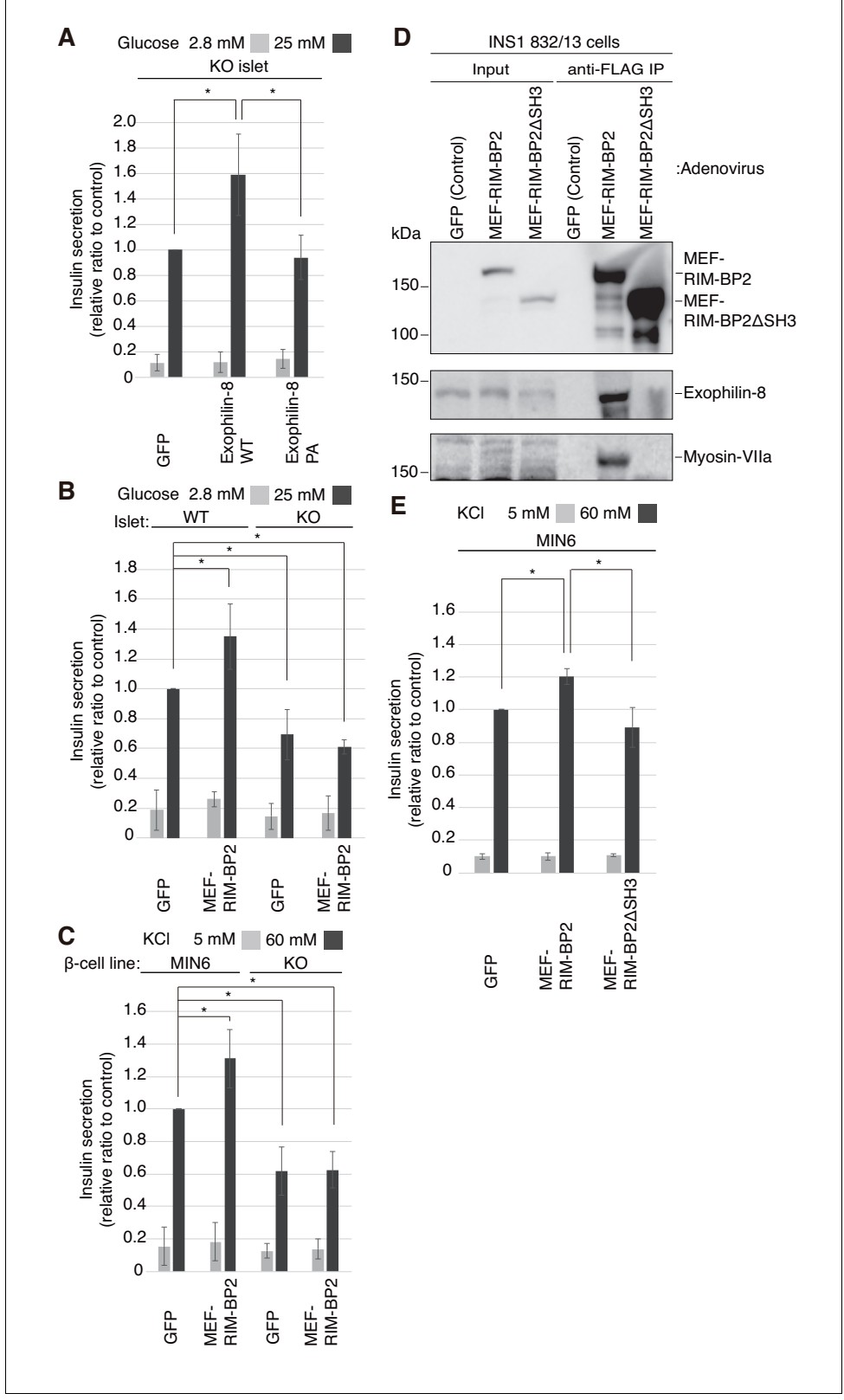

**Figure 6.** Exophilin-8 and RIM-BP2 potentiate insulin secretion through their mutual interaction. (**A**) Pancreatic islets isolated from exophilin-8-knockout (KO) mice were infected with adenovirus encoding control GFP or MEF-tagged, wild-type (WT) or PA (ABD) mutant exophilin-8, as shown in *Figure 6—figure supplement 1A,B*, and were cultured in a fresh medium for 48 hr followed by Krebs-Ringer bicarbonate (KRB) buffer containing 2.8 mM
*Figure 6 continued on next page*

*Figure 6 continued*

glucose for 30 min. The cells were then incubated in the same buffer for 30 min (gray bars) followed by the buffer containing 25 mM glucose for 30 min (black bars). The ratios of insulin secreted into the medium to that remaining in the cells were normalized to those found in control cells stimulated by the secretagogue. (B, C) WT or KO islets (B), and MIN6 or exophilin-8-null β-cell lines (C) were infected with adenovirus encoding either GFP or MEF-RIM-BP2, as shown in *Figure 6—figure supplement 1C and D*, respectively, and were subjected to insulin secretion assays as in (A) stimulated by either 25 mM glucose (B) or 60 mM KCl (C) for 30 min. (D) INS-1 832/13 cells were infected with adenovirus encoding control GFP, wild-type MEF-RIM-BP2, or MEF-RIM-BP2ΔSH3 lacking all the SH3 domains, and the immunoprecipitates with anti-FLAG antibody, as well as 1/100 of the original extracts, were immunoblotted with anti-FLAG, anti-exophilin-8, and anti-myosin-VIIa antibodies. Note that RIM-BP2ΔSH3 lost binding activities to exophilin-8 and myosin-VIIa. (E) MIN6 cells were infected with adenovirus encoding control GFP, or wild-type or ΔSH3 RIM-BP2, as shown in *Figure 6—figure supplement 1E*, and were subjected to insulin secretion assays as in (C). All quantitative data are means ± SD ($n = 3$). *p values calculated using two-tailed unpaired *t*-test are as follows: (A) 0.03278 (GFP), 0.03607 (Exophilin-8 PA) vs Exophilin-8 WT, (B) 0.04952 (WT-islet MEF-RIM-BP2), 0.03503 (KO-islet GFP), 0.00015 (KO-islet MEF-RIM-BP2) vs WT-islet GFP, (C) $9.4 \times 10^{-5}$ (MIN6 MEF-RIM-BP2), $2.5 \times 10^{-6}$ (KO GFP), $1.3 \times 10^{-7}$ (KO MEF-RIM-BP2) vs MIN6 GFP, and (E) $1.7 \times 10^{-3}$ (MIN6 GFP), 0.014 (MIN6 MEF-RIM-BP2ΔSH3) vs MIN6 GFP MEF-RIM-BP2.

The following figure supplement is available for figure 6:

**Figure supplement 1.** Expression levels of exophilin-8 and RIM-BP2.

is ~170 kDa (*Figure 8B,C*). As described in *Figure 3A*, myosin-VIIa was identified with RIM-BP2 from exophilin-8-interacting, 150 ~ 190 kDa protein bands in MIN6 cells by LC-MS/MS. Furthermore, in INS-1 832–13 cells, both minor 260 kDa and major 170 kDa protein bands were similarly downregulated by myosin-VIIa siRNA treatment (*Figure 9C*). To further examine whether the 170 kDa protein represents myosin-VIIa specifically expressed in β-cells, we examined RIM-BP2-binding myosin-VIIa in INS1 832/13 cells expressing OST-tagged RIM-BP2, using another myosin-VIIa antibody we generated. We found that only the 170 kDa protein was pulled down with RIM-BP2, although the 260 kDa protein was also pulled down after exogenous expression of myosin-VIIa (*Figure 9D*). These findings indicate that the 170 kDa protein is derived from myosin-VIIa endogenously expressed in β-cells. By contrast, the 260 kDa protein is derived from transfected myosin-VIIa cDNA and interacts with RIM-BP2 through exophilin-8, because it can directly interact with exophilin-8 but not with RIM-BP2 (*Figure 9A,B*). Finally, we investigated the endogenous complex formation in INS1 832/13 cells by sucrose density gradient centrifugation (*Figure 9—figure supplement 1*). Exophilin-8 (130 kDa) and RIM-BP2 (150 kDa) exhibited similarly wide distributions in fractions 3–10, whereas both endogenous 170 kDa and 260 kDa myosin-VIIa showed relatively narrower distributions around fractions 6–10. Judged from the positions of molecular mass markers, these findings are consistent with the ternary complex formation among these proteins in the cells, although each protein also appeared to exist as a monomer.

## Exophilin-8 anchors granules to the actin cortex through RIM-BP2 and myosin-VIIa

We then investigated the silencing effects of the exophilin-8-interacting proteins. In pancreatic β-cells, the two isoforms of RIM, RIM1 and RIM2, are expressed (*Iezzi et al., 2000*; *Yasuda et al., 2010*), and homozygous ablation of RIM2 or RIM-interacting Munc13-1 has been shown to affect insulin secretion profoundly (*Kang et al., 2006*; *Yasuda et al., 2010*). The L-type $Ca_v1.3$ subtype is a predominant VDCC mediating stimulus-induced $Ca^{2+}$ influx to trigger insulin secretion (*Yang and Berggren, 2006*). Consistently, when these proteins were downregulated separately by specific siRNA (*Figure 10—figure supplement 1*), glucose-induced insulin secretion was markedly decreased (*Figure 10A*). However, the peripheral accumulation of granules was not affected (*Figure 10B,C*, *Figure 10—figure supplement 2*), suggesting that these proteins function after granules are recruited to the cell periphery. Endogenous myosin-Va and -VIIa and their respective C-terminal tails expressed exogenously in INS-1 832/13 cells were all associated with insulin granules, including those accumulated in the actin-rich cell periphery (*Figure 10D*, *Figure 10—figure supplement 3A,B*). Although silencing of either myosin-Va or -VIIa decreased insulin secretion, only

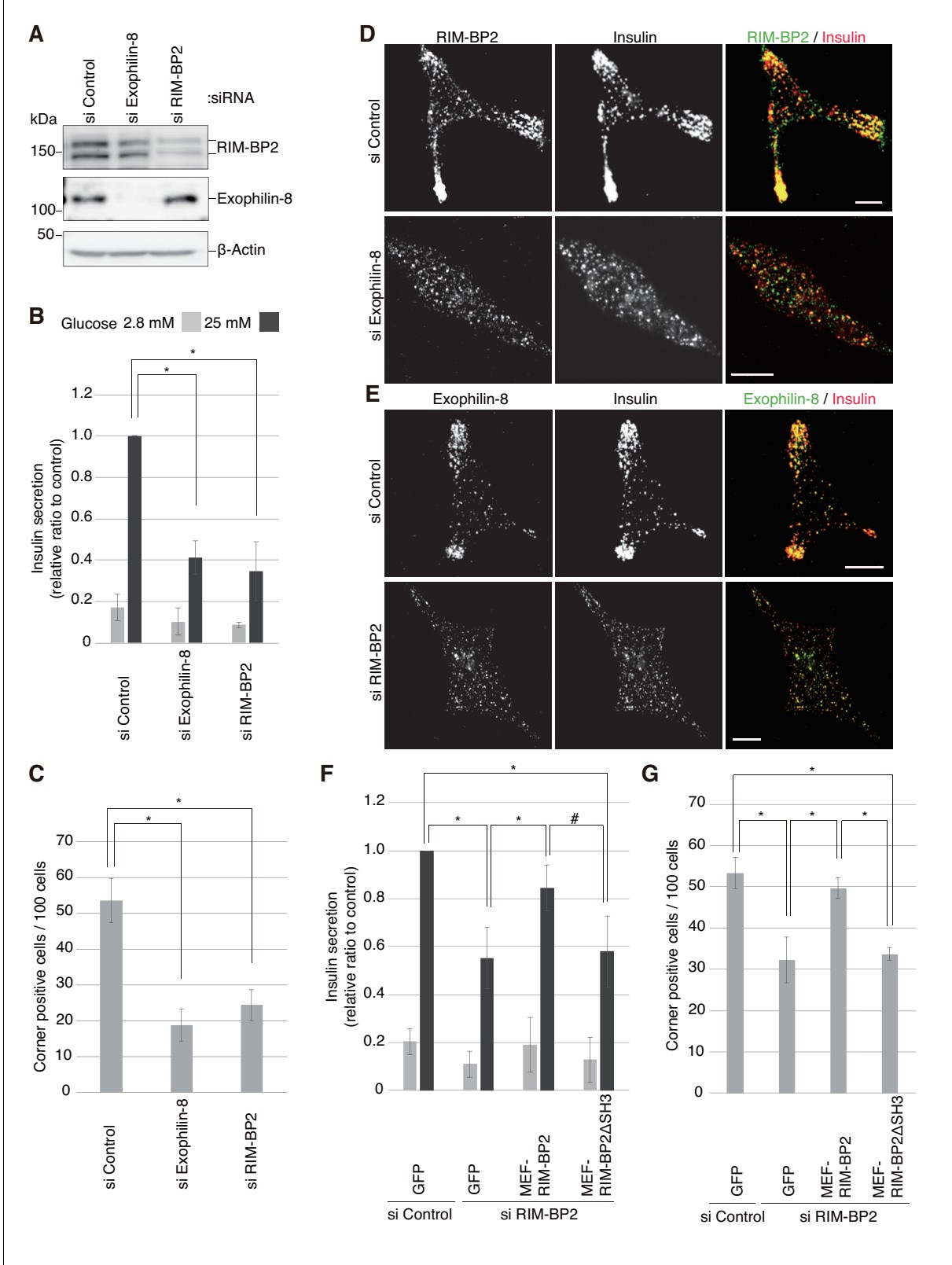

**Figure 7.** Silencing of exophilin-8 or RIM-BP2 decreases the cell-corner localization and exocytosis of insulin granules. (**A**) INS-1 832/13 cells were transfected with control siRNA duplexes, or siRNA duplexes against exophilin-8 or RIM-BP2. The cell extracts were immunoblotted with anti- RIM-BP2, anti-exophilin-8, and anti-$\beta$-actin antibodies. (**B**) INS-1 832/13 cells treated with siRNA as shown in (**A**) were incubated for 2 hr in KRB buffer containing 2.8 mM glucose, and were then stimulated for 60 min in the same buffer (gray bars) or buffer containing 25 mM glucose (black bars). The ratios of

*Figure 7 continued on next page*

*Figure 7 continued*

insulin secreted into the medium to that remaining in the cells were normalized to those found in control cells stimulated by 25 mM glucose. (C–E) The cells treated with siRNA were immunostained with anti-insulin antibody and either anti-RIM-BP2 (D) or anti-exophilin-8 (E) antibodies. In each experiment, a total of 100 cells were visually inspected, and the fraction of cells exhibiting a higher granule density in at least one cell corner or extension than in the cell center was manually counted (C). Bars, 10 μm. (F, G) INS-1 832/13 cells treated with control siRNA or siRNA against RIM-BP2 were infected with adenovirus encoding control GFP, or wild-type or ΔSH3 MEF-RIM-BP2, as shown in *Figure 7—figure supplement 2*. They were subjected to insulin secretion assays (F) as in (B), or were immunostained with anti-insulin antibody to examine granule localization (G) as in (C). All quantitative data are means ± SD ($n$ = 5 for B, and $n$ = 4 for C, and $n$ = 3 for F). *p values calculated using two-tailed unpaired $t$-test are as follows: (B) $1.9 \times 10^{-7}$ (si Exophilin-8), $7.3 \times 10^{-6}$ (si RIM-BP2) vs si Control, (C) 0.00010 (si Exophilin-8), 0.00025 (si RIM-BP2) vs si Control, (F) $3.7 \times 10^{-3}$ (si RIM-BP2, GFP), $8.2 \times 10^{-3}$ (si RIM-BP2, MEF-RIM-BP2) vs si Control, GFP, and 0.033 (si RIM-BP2, GFP), 0.060 (si RIM-BP2, MEF-RIM-BP2ΔSH3; marked by #) vs si RIM-BP2, MEF-RIM-BP2, and (G) $1.1 \times 10^{-3}$ (si RIM-BP2, GFP), $5.5 \times 10^{-3}$ (si RIM-BP2, MEF-RIM-BP2) vs si Control, GFP, and $7.7 \times 10^{-3}$ (si RIM-BP2, GFP), $7.1 \times 10^{-5}$ (si RIM-BP2, MEF-RIM-BP2ΔSH3) vs si RIM-BP2, MEF-RIM-BP2.

The following figure supplements are available for figure 7:

**Figure supplement 1.** More images of INS-1 832/13 cells treated with siRNA.

**Figure supplement 2.** Expression levels of endogenous and exogenous RIM-BP2.

**Figure supplement 3.** Granule localization in INS-1 832/13 cells treated with siRNA against RIM-BP2 and then expressing wild-type or ΔSH3 RIM-BP2.

the latter affected the peripheral accumulation of granules (*Figure 10A–C*, *Figure 10—figure supplement 2*), suggesting that the two motor proteins may have differential effects on secretory granule positioning.

## Discussion

In the present study, we showed that mice deficient in the Rab27a effector, exophilin-8, exhibit glucose intolerance and impaired insulin secretion in vivo. We further identified a new multiprotein complex, in which exophilin-8 directly interacts with RIM-BP2 and then binds the actin-motor protein, myosin-VIIa, and the exocytic machinery, such as Ca$_v$1.3 and RIM. Disruption of the interaction by ablation or mutation of either exophilin-8 or RIM-BP2 leads to loss of polarized granule distribution and to a marked decrease in granule exocytosis. RIM-BP2 also has binding activities to the α$_1$ subunits of L-, N-, and P/Q type VDCC (*Hibino et al., 2002*). On the other hand, RIM, originally identified as an effector of Rab3 (*Wang et al., 1997*), binds and activates a priming factor, Munc13-1 (*Deng et al., 2011*), and also has binding activities to N- and P/Q type VDCC, but not to L-type VDCC (*Kaeser et al., 2011*). Because RIM binds Rab3, but not Rab27 (*Fukuda, 2003*), and because exophilin-8 binds Rab27, but not Rab3 (*Fukuda and Kuroda, 2002*), the RIM-BP2–myosin-VIIa complex can theoretically associate with granules via either Rab27a–exophilin-8 or Rab3–RIM (*Figure 9E*).

Although an interaction of exophilin-8 with myosin-Va has been shown in heterologous cells and in vitro (*Fukuda and Kuroda, 2002*; *Desnos et al., 2003*), we cannot find evidence that the two proteins act together in secretory cells: they do not form a complex and silencing of myosin-Va does not affect peripheral granule accumulation unlike that of exophilin-8, although it is possible that the degree of myosin-Va depletion in our experimental condition is enough to decrease insulin secretion, but not to affect granule localization. Some question remains regarding the physiological and functional interaction between exophilin-8 and myosin-Va. For example, exophilin-8 does not interact with myosin-Va under physiological conditions in MIN6 cells, where only the brain-isoform of myosin-Va is expressed (*Brozzi et al., 2012*). Furthermore, these two proteins show negligible physical and functional interactions in the melanocyte cell line, Melan-a, where the melanocyte-isoform of myosin-Va is expressed (*Kuroda and Fukuda, 2005*). We instead show that myosin-VIIa associates with secretory granules and indirectly interacts with exophilin-8 via RIM-BP2. Furthermore, silencing of myosin-VIIa disrupts granule clustering in the actin-rich cell periphery and markedly decreases stimulus-induced granule exocytosis, as does the silencing of exophilin-8. Myosin-VIIa appears to function in the anchoring, rather than in the active transport, of granules within the F-actin network,

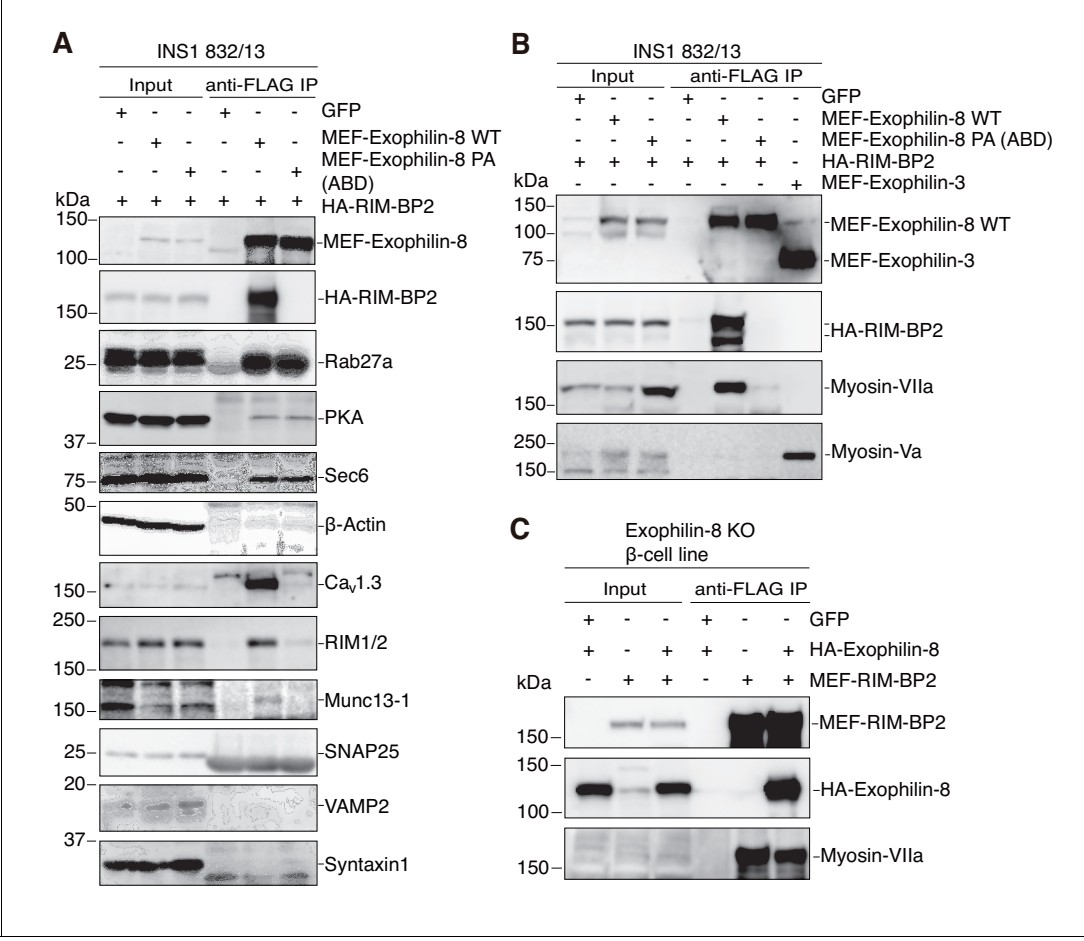

**Figure 8.** Exophilin-8 indirectly interacts with myosin-VIIa via RIM-BP2 in INS-1 832/13 cells. (**A, B**) INS-1 832/13 cells were infected with adenoviruses expressing HA-RIM-BP2 and either control GFP, or MEF-tagged, wild-type (WT) or PA (ABD) mutant exophilin-8 (**A, B**), and with that expressing MEF-exophilin-3 (**B**). The immunoprecipitates with anti-FLAG antibody, as well as 1/100 of the original extracts, were immunoblotted with the indicated antibodies to investigate the interacting proteins. The protein bands found in Input lanes of Munc13-1 are non-specific proteins, whereas those found in the immunoprecipitates of SNAP25 are immunoglobulin G. (**C**) Exophilin-8-null β-cell lines were infected with adenoviruses expressing GFP, HA-tagged wild-type exophilin-8, and/or MEF-RIM-BP2. The immunoprecipitates with anti-FLAG antibody were immunoblotted with the indicated antibodies as in (**B**).

because exophilin-8-positive granules are markedly immobile in the resting state in living cells (*Desnos et al., 2003*; *Mizuno et al., 2011*; *Huet et al., 2012*).

It should be noted that myosin-VIIa identified in the exophilin-8–RIM-BP2 complex in β-cells shows a molecular mass ~170 kDa in gels, much smaller than its authentic mass ~260 kDa. Because this form does not appear to be generated from transfected myosin-VIIa cDNA in INS1 832/13 cells, it may represent an alternatively spliced form that specifically functions in secretory cells. In fact, such a small myosin-VIIa isoform has been reported in the UCSC Genome Browser (https://genome.ucsc.edu/). It has recently been shown that, among the four alternatively spliced isoforms of the minus-ended motor, myosin-VIa, the small insert isoform specifically tethers secretory granules to the cortical actin (*Tomatis et al., 2013*), although its molecular link to granules, such as Rab and its effector, has not been identified. Although the exact molecular nature of the 170 kDa form of myosin-VIIa and its interaction mode with RIM-BP2 are unknown, our findings that either the PA (ABD) exophilin-8 or the ΔSH3 RIM-BP2 fails to restore the peripheral granule accumulation or to rescue the decreased insulin secretion in knockdown or knockout cells strongly indicate that the exophilin-8–RIM-BP2–170 kDa–myosin-VIIa complex formation is functionally critical for granule exocytosis. Currently, roles for myosin-VIIa are well-documented in actin-rich protrusion, such as stereocilia of

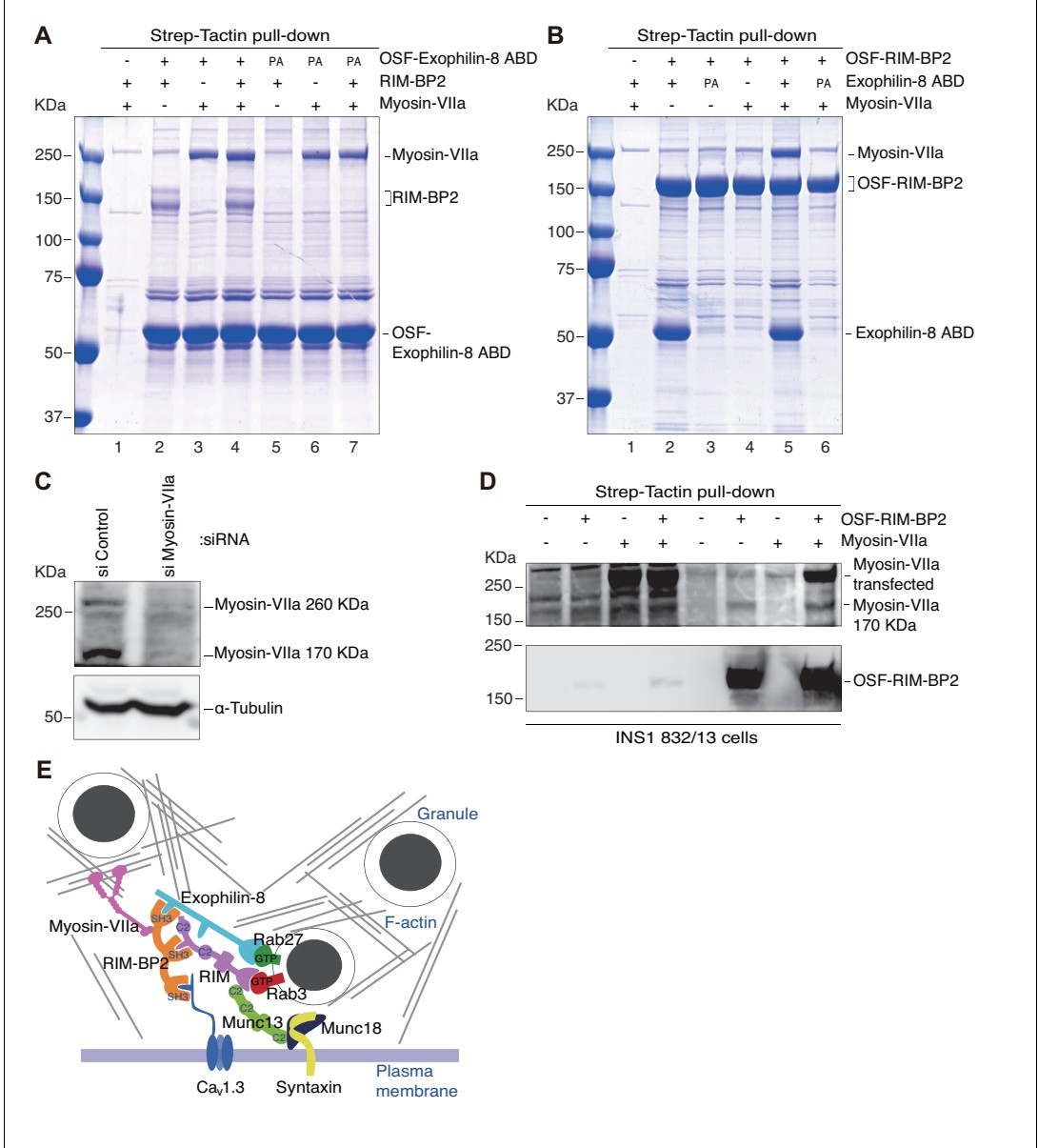

**Figure 9.** The molecular mass of myosin-VIIa interacted with RIM-BP2 in INS-1 832/13 cells is 170 kDa, whereas that directly binds exophilin-8 in HEK293A cells is 260 kDa. (**A**) HEK293A cells were transfected with plasmids expressing RIM-BP2, myosin-VIIa, and/or OSF-tagged, wild-type or PA mutant exophilin-8 ABD. Exophilin-8 ABD and the binding proteins were pulled down using Strep-Tactin beads, and were subjected to SDS-PAGE and Coomassie Brilliant Blue staining. (**B**) HEK293A cells were transfected with plasmids expressing wild-type or PA mutant exophilin-8 ABD, myosin-VIIa, and OSF-tagged RIM-BP2. RIM-BP2 and the binding proteins were pulled down as in (**A**). (**C**) INS-1 832/13 cells were transfected with control siRNA duplexes or siRNA duplexes against myosin-VIIa, and the cell extracts were immunoblotted with anti-myosin-VIIa and anti-α-tubulin antibodies. Note that both 260 kDa and major 170 kDa protein bands were downregulated by myosin-VIIa siRNA. (**D**) INS-1 832–13 cells were transfected with plasmids expressing myosin-VIIa and/or OSF-tagged RIM-BP2. Because the transfection efficiency in these cells are poor compared with that in HEK293A cells, the binding proteins were detected by immunoblotting. The anti-myosin-VIIa used in this figure was that generated by our laboratory (αMyo7), in contrast to the commercially available antibody used in *Figures 6D*, *8*, *9C* and *10* (see Materials and methods). (**E**) Illustration of the cortical F-actin network with secretory granules and the plasma membrane, the exophilin-8–RIM-BP2–170 kDa myosin-VIIa complex found in the present study, and the previously known, exocytic protein interactions (*Südhof, 2013*). The exact molecular nature of the 170 kDa form of myosin-VIIa and whether it directly interacts with RIM-BP2 are currently unknown.

The following figure supplement is available for figure 9:

**Figure supplement 1.** Sucrose density gradient centrifugation analysis of the exophilin-8–RIM-BP2–myosin-VIIa complex.

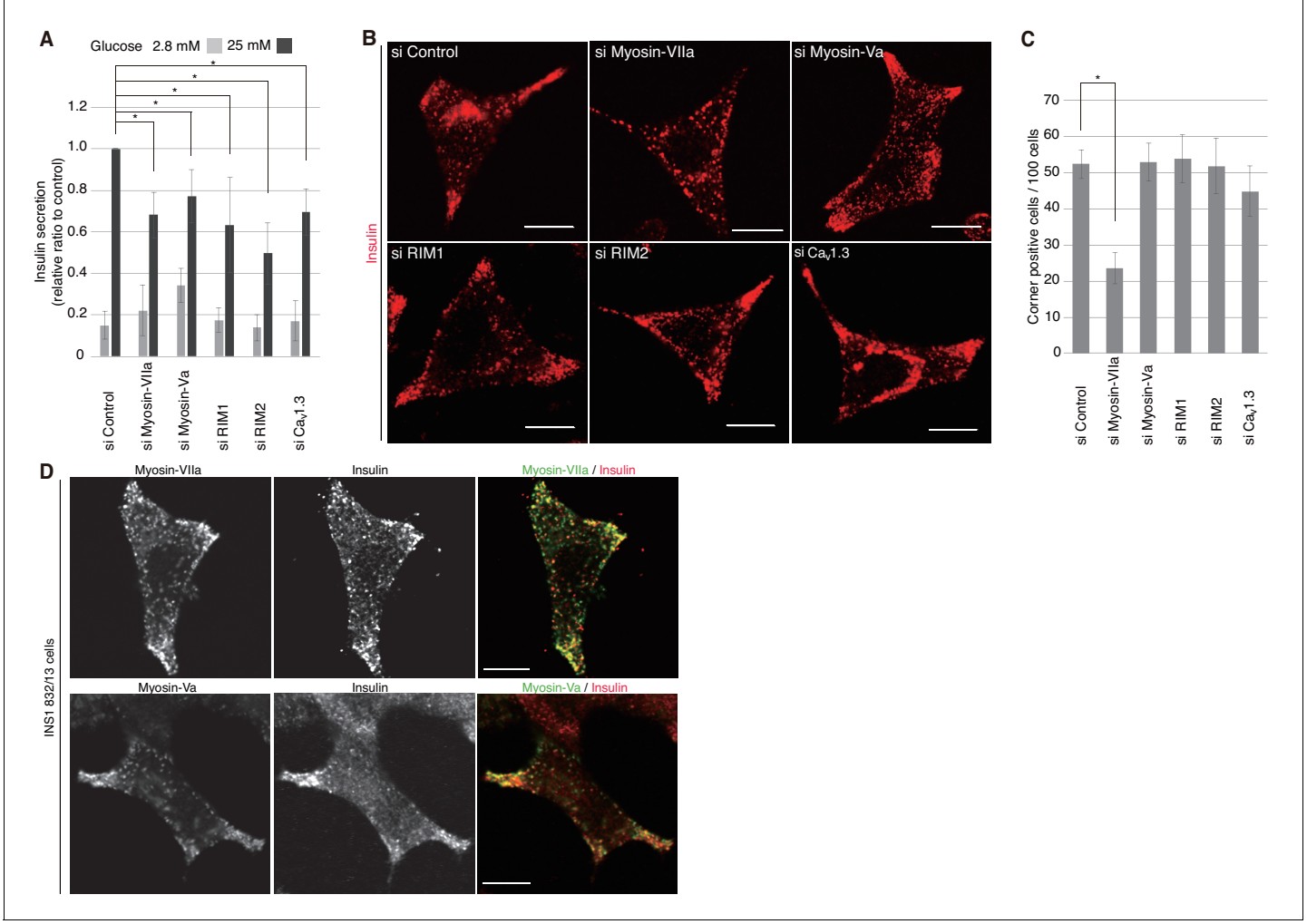

**Figure 10.** Myosin-VIIa clusters granules at cell corners. (A–C) INS-1 832–13 cells were transfected with control siRNA duplexes or siRNA duplexes against the indicated proteins (*Figure 9B*, *Figure 10—figure supplement 1*) and were subjected to insulin secretion assays (A) as described in *Figure 7B*, or to insulin immunostaining (B, C) to examine the peripheral accumulation of granules as described in *Figure 6C–E*. (D) INS-1 832/13 cells were coimmunostained with anti-insulin antibody and either with anti-myosin-Va or -VIIa antibody. Bars, 10 μm. All quantitative data are means ± SD ($n$ = 4). *p values calculated using two-tailed unpaired $t$-test are as follows: (A) 0.00029 (si Myosin-VIIa), 0.01168 (si Myosin-Va), 0.01881 (si RIM1), 0.00688 (si RIM2), 0.00045 (si $Ca_v$1.3) vs si Control, and (B) $6.2 \times 10^{-5}$ (si Myosin-VIIa) vs si Control.

The following figure supplements are available for figure 10:

**Figure supplement 1.** Silencing of myosin-Va, RIM, and $Ca_v$1.3 by siRNA.

**Figure supplement 2.** More images of INS-1 832–13 cells treated with siRNA.

**Figure supplement 3.** Localization of F-actin and its motor proteins.

cochlear and vestibular hair cells and microvilli in intestine brush borders and photoreceptor cells (*Yan and Liu, 2010*; *Lu et al., 2014*), but no such roles have been identified in granule anchoring or trafficking. This may be because of the existence of parallel secretory pathways bypassing the complex formation: all the granules may not be captured within the cortical F-actin network, or some granules may be captured there by other mechanisms. In fact, exophilin-8-null islets preserve approximately half of the normal level insulin secretion in response to glucose or depolarization stimulation. Further research is required to elucidate the mechanism by which secretory cells differentially or redundantly engage multiple motor proteins, such as myosins-Va, -VI, and -VIIa.

Under electron microscopy, exophilin-8-null pancreatic $\beta$-cells exhibit a decrease in the number of granules close to the plasma membrane, but still harbor docked granules whose limiting membrane is directly attached to the membrane. Using the same experimental protocol, we previously demonstrated that $\beta$-cells deficient in another Rab27 effector, granuphilin, exhibit a much greater decrease in the number of granules close to the plasma membrane, and almost complete loss of docked granules (*Gomi et al., 2005*). It is notable that the granule number decrease in granuphilin-null $\beta$-cells occurs in a more restricted area, with granule centers residing within 200 nm of the plasma membrane, whereas the centers in exophilin-8-null $\beta$-cells are within 300 nm. This difference has a significant effect on the number of the granules directly attached to the plasma membrane, given that the diameter of insulin granules is 300 ~ 350 nm. Furthermore, when cultured $\beta$-cells over-expressing these proteins are observed under optical microscopy, granuphilin redistributes granules along the plasma membrane (*Torii et al., 2004*), whereas exophilin-8 does so in a relatively broad peripheral area (*Mizuno et al., 2011*). Therefore, it is likely that only a portion of exophilin-8 and its associated granules are linked to the $Ca^{2+}$ channel and the exocytic machinery along the plasma membrane. These findings indicate that, compared with granuphilin, exophilin-8 plays a relatively minor role in granule docking, although it may indirectly contribute to the process by recruiting granules close to the plasma membrane.

What is the physiological relevance of the exophilin-8-induced peripheral granule accumulation found in monolayer $\beta$-cells? Although pancreatic $\beta$-cells do not display the typical epithelial polarity, they concentrate F-actin and exocytic machinery, including $Ca_v1.3$, at cell edges and particularly at cell vertexes facing blood vessels within islets (*Geron et al., 2015*). This polarity seems to be functionally important, because exocytosis in islets preferentially occurs in the vicinity of vessels, around which $\beta$-cells form rosettes (*Takahashi et al., 2002*). The exophilin-8–RIM-BP2–myosin-VIIa complex can provide the molecular basis of these observations, because it interacts with both F-actin and $Ca_v1.3$. In fact, we found that loss of exophilin-8 prevents polarized granule accumulation both in $\beta$-cells within islets and in those cultured in a monolayer. Therefore, even monolayer $\beta$-cells could display such a polarity in a cell-autonomous manner. Monolayer chromaffin cells have also been shown to cluster L- and P/Q type $Ca^{2+}$ channels and exocytic machinery on the border of cortical F-actin cages (*Torregrosa-Hetland et al., 2011*). How, then, do exophilin-8-positive granules captured within the F-actin network fuse upon secretagogue stimulation? It has recently been shown that the cortical actomyosin II network acts like a casting net to drive granules towards the plasma membrane by undergoing relaxation in a secretagogue-dependent manner (*Papadopulos et al., 2015*). Thus, the stimulus-induced $Ca^{2+}$ influx not only triggers a final fusion reaction but also may help granules within the F-actin network make further access to the exocytic site.

In summary, the novel exophilin-8–RIM-BP2–myosin-VIIa complex seems to replenish and to maintain a releasable pool of granules beneath the plasma membrane by locating granules both within the peripheral F-actin network and close to the $Ca^{2+}$ channel and exocytic machinery. This may be the first to provide the molecular basis correlating the physical location of secretory granules with their functional pool, given that stable docking to the plasma membrane does not necessarily form a releasable granule pool (*Gomi et al., 2005*; *Mizuno et al., 2016*). Genetic or functional alterations of the complex may lead to type 2 diabetes and other secretory disorders in humans.

## Materials and methods

### Generation of exophilin-8-deficient mice

The exophilin-8 (encoded by the *Myrip* gene) knockout mice (Accession No. CDB1100K: http://www2.clst.riken.jp/arg/mutant%20mice%20list.html) were generated as described elsewhere (http://www2.clst.riken.jp/arg/Methods.html). To construct a targeting vector, genomic fragments of the exophilin-8 locus were obtained from a BAC clone, RP24-276O9 (BACPAC Resources). The exon4 of the exophilin-8 gene was disrupted by insertion of a loxP-flanked cassette of the neomycin resistance gene under the *pgk* promoter. Targeted TT2 (derived from F1 of C57BL/6 and CBA) embryonic stem cell clones (*Yagi et al., 1993*) were microinjected into eight-cell stage ICR embryos, and were then transferred into pseudopregnant ICR females. The resulting chimeras were bred with C57BL/6 mice, and heterozygous offspring were identified by Southern blotting and polymerase chain reaction (PCR). The primers used for PCR were *Myrip/Fow1* (5′-GATGGGTCCTGCTTCTCACC-

3') and *Myrip/Rev1* (5'-CTCCGCCCTCTTTCCAGAAC-3') for the wild-type allele, and *Neo/Fow2* (5'-AGGACATAGCGTTGGCTACC-3) and *Myrip/Rev1* for the targeted allele. Mutant lines were backcrossed with C57BL/6 mice. The protocol for animal experimentation was approved by the Institutional Animal Care and Use Committee of RIKEN Kobe Branch. All the animal experiments were conducted in accordance with the RIKEN institutional guideline and the rules and regulations of the Animal Care and Experimentation Committee, Gunma University.

## Phenotypic analyses of mice

Wild-type and knockout males for the experiments were obtained by heterozygous mating between backcrossed progenies. Mice had free access to water and standard laboratory chow in an air-conditioned room with a 12 hr light/12 hr dark cycle. An intra-peritoneal glucose tolerance test (1 g glucose/kg body weight) and an intra-peritoneal insulin tolerance test (0.75 U human insulin/kg body weight) were performed as described previously (*Wang et al., 2013*). Blood glucose levels were determined by a glucose oxidase method using Glutest sensor and Glutest Pro GT-1660 (Sanwa Kagaku Kenkyujyo, Nagoya, Japan). Islet isolation by pancreatic duct injection of collagenase solution and insulin secretion assay in perifused islets were performed as described previously (*Wang et al., 2013*). Briefly, isolated islets were perifused with standard low-glucose (2.8 mM) Krebs-Ringer bicarbonate (KRB) buffer for 30 min. Thereafter, the collection of each fraction (1 ml/min) was started, and an appropriate secretagogue was applied at 10 min after the start. Insulin was measured using an AlphaLISA insulin kit with an EnVision 2101 Multilabel Reader (PerkinElmer, Waltham, MA). The examination of granule distribution by electron microscopy was performed as described previously (*Gomi et al., 2005*).

## DNA construction

Mouse exophilin-8 cDNA (*Mizuno et al., 2011*) was subcloned into the pcDNA3-MEF, pcDNA3-FLAG, and pENTR3C-MEF vector (*Ichimura et al., 2005*). Site-directed mutagenesis of exophilin-8 was performed using the following primers: 5'-CTGCAG<u>GCG</u>AAGGCC<u>GCT</u>AAGAAC<u>GCT</u>GCAGTG-3' and 5'-CACTGC<u>AGC</u>GTTCTT<u>AGC</u>GGCCTT<u>CGC</u>CTGCAG-3' for PA (MBD) and 5'-CAGAGG<u>GC-G</u>AAACTG<u>GCT</u>GCCCCT<u>GCT</u>GTGAAA-3' and 5'-TTTCAC<u>AGC</u>AGGGGC<u>AGC</u>-CAGTTT<u>CGC</u>CCTCTG-3' for PA (ABD). The resulting PA (MBD) and PA (ABD) mutants carry substitutions of alanine for arginine and proline at amino acid positions 474, 477, 480, and 798, 801, 804, respectively. The cDNA fragment encoding 1–167 amino acids of exophilin-8 was subcloned into the pGEX4T-1 (GE Healthcare, Little Chalfont, UK) and pMAL-CRI (NEB) to generate glutathione S-transferase (GST)-fused and maltose-binding protein (MBP)-fused proteins, respectively. Mouse RIM-BP2 and myosin-VIIa cDNAs were amplified from the cDNA of MIN6 cells. Mouse wild-type RIM-BP2 resistant to siRNA 11 toward rat RIM-BP2 (GE Dharmacon; Lafayette, CO) was generated by site-directed mutagenesis using the following primers: 5'-GCGAATTCATGCGAGAGGCTGCTGAGC-3' and 5'-AGGAGAACCAAGGCACTGATC-3', and 5'-CCACTCGAGCTAAGGGGGCTGGCTTACCC-3' and 5'-GATCAGTGCCTTGGTTCTCCT-3'. Deletion mutant of RIM-BP2ΔSH3 was made from the above siRNA-resistant RIM-BP2 using the primers: 5'-GCGAATTCATGCGAGAGGCTGCTGAGC-3' and 5'-CCACTCGAGCTACTCCTCAGCACCAGGGTC-3', and 5'-TCCAAGCAAAGCAGCTCGAATGAGTCGCGGCTGGCT-3' and 5'-AGCCAGCCGCGACTCATTCGAGCTGCTTTGCTTGGA-3'. The full-length and deleted cDNAs were subcloned into the pcDNA3-3×HA, EGFP-C2 (BD Biosciences, Franklin Lakes, NJ), pENTR3C-MEF, and pENTR3C-3×HA. They were also subcloned into the pCAG vector with or without an OSF tag (*Morita et al., 2007*; *Matsunaga et al., 2017*). Recombinant adenoviruses were prepared as described previously (*Wang et al., 2013*).

## Antibodies, immunoblotting, and immunoprecipitation

Rabbit anti-exophilin-8 antibody (αExo8N) was raised against GST-fused N-terminal exophilin-8 protein (1–167 amino acids). The sera were passed through a column containing the same N-terminal exophilin-8 protein fused with MBP. The affinity-purified antibodies were then eluted and concentrated. Rabbit anti-myosin-VIIa antibody (αMyo7) was raised against MBP-fused, mouse myosin-VIIa protein (1560–1727 amino acids). Mouse anti-myc 9E10 monoclonal antibody was purified from the ascites fluid of a hybridoma-injected mouse. Guinea pig anti-porcine insulin serum was a gift from H. Kobayashi (Gunma University). The following commercially purchased antibodies were also used:

rabbit polyclonal antibodies toward FLAG (F7425, Sigma-Aldrich, St. Louis, MO; RRID:AB_439687), myosin-Va (LF-18, Sigma-Aldrich; RRID:AB_260545), RIM-BP2 (15716–1-AP, Proteintech, Rosemont, IL), Munc13-1 (55053–1-AP, Proteintech), HA (561, MBL, Nagoya, Japan; RRID:AB_591839), green fluorescent protein (GFP; 598; MBL; RRID:AB_2313843), Rab27a/b (18975; IBL, Fujioka, Japan; RRID: AB_494635), PKA (ab26322; Abcam, Cambridge, United Kingdom), myosin-VIIa (ab3481; Abcam; RRID:AB_303841), $Ca_v1.3$ (ACC-005; Alomone Labs, Jerusalem, Israel; RRID:AB_2039775), RIM1/2 (140203; Synaptic Systems, Goettingen, Germany), and VAMP2 (627724; Calbiochem, San Diego, CA; RRID:AB_212589); and mouse monoclonal antibodies toward glyceraldehyde-3-phosphate dehydrogenase (GAPDH; 3H12; MBL), α-tubulin (T5168; Sigma-Aldrich; RRID:AB_477579), β-actin (A5316; Sigma-Aldrich; RRID:AB_476743), syntaxin1 (HPC-1; Sigma-Aldrich; RRID:AB_592786), $Na^+$-$K^+$ ATPase (C464.6; Upstate Biotechnology, Lake Placid, NY; RRID:AB_309699), Sec6 (ADI-VAM-SV021; Assay Designs, Ann Arbor, MI), and SNAP25 (610366; BD Biosciences; RRID:AB_397752) Tissue extract preparation, immunoblotting, and immunoprecipitation were performed as described previously (*Matsunaga et al., 2017*). Each image resulting from immunoblotting is representative of at least three independent experiments.

## Cell culture, transfection, and adenovirus infection

All cells were cultured in a humidified incubator with 95% air and 5% $CO_2$ at 37°C. MIN6 cells (originally provided from Dr. Jun-ichi Miyazaki, Osaka University; *Miyazaki et al., 1990*) were cultured in Dulbecco's modified Eagle's medium (DMEM) containing 15% fetal bovine serum (FBS) supplemented with 1 mM L-glutamine and 50 μM 2-mercapthoethanol. INS-1 832/13 cells (originally provided from Dr. Christopher Newgard, Duke University; *Hohmeier et al., 2000*) were cultured in RPMI1640 containing 10% FBS supplemented with 1 mM L-glutamine, 1 mM HEPES, 1 mM sodium pyruvate, and 50 μM 2-mercapthoethanol. HEK293A cells (Invitrogen, Carlsbad, CA) were cultured in DMEM containing 10% FBS supplemented with 1 mM L-glutamine. INS1 832/13 (RRID: CVCL_7226), MIN6 (RRID: CVCL_0431), and HEK293A (RRID: CVCL_6910) cells were listed by NCBI Bio sample (BioSample: SAMEA4104055, SAMEA4168040, and SAMEA4146837, respectively). Exophilin-8-null β-cell lines were established from exophilin-8-null mice, by a method similar to that by which granuphilin-null β-cell lines were previously established (*Mizuno et al., 2016*). These cell lines have tested for mycoplasma contamination by 4',6-diamidino-2-phenylindole staining. Plasmid transfections and adenovirus infections were performed as described previously (*Wang et al., 2013*).

## MEF tag-based protein purification and tandem mass spectrometry

The purification procedure was similar to that reported previously (*Ichimura et al., 2005*), with minor modifications. Briefly, the extracts of MIN6 cells expressing MEF-exophilin-8 were subjected to immunoprecipitation with anti-myc antibody, cleavage by TEV protease, immunoprecipitation with an anti-FLAG antibody, and FLAG peptide-dependent elution. The final eluate was separated by SDS-PAGE and visualized by Oriole fluorescent gel staining (BioRad, Hercules, CA). Specific bands were excised and digested in the gel with trypsin, and the resulting peptide mixtures were analyzed by nanoflow LC-MS/MS at Gunma University. All MS/MS spectra were searched against the non-redundant RefSeq protein sequence database at the National Center for Biotechnology Information using Mascot software (Matrix Science, London, UK).

## Immunofluorescence microscopy

INS-1 832/13 cells cultured on coverslips were fixed with 3% paraformaldehyde in phosphate buffered saline (PBS) for 30 min and permeabilized with 0.1% Triton X-100 in PBS for 30 min. Isolated pancreatic islets were cultured on 35-mm glass bottom dishes for 3 days, fixed with 3% paraformaldehyde in PBS for 60 min, and permeabilized with 0.2% Triton X-100 in PBS for 60 min. The cells or islets were then treated with 50 mM $NH_4Cl$-PBS for 10 min at room temperature and blocked with PBS containing 1% bovine serum albumin for 15 min. They were incubated with primary antibodies (diluted at 1:100 or 1:200) for 2 hr or overnight, washed three times with PBS, and then incubated with Alexa Fluor 488- or 568-conjugated secondary antibodies (Invitrogen; diluted at 1:500) for 60 min. Finally, INS-1 832/13 cells were mounted using Slow Fade Gold reagent (Invitrogen), whereas the islets were filled with PBS. Both samples were observed using laser scanning confocal microscopes, A1 (Nikon, Tokyo, Japan) equipped with a 100× oil immersion objective lens (1.49 NA) or

FV1000 (Olympus, Tokyo, Japan) equipped with 100× oil immersion objective lens (1.40 NA). The images were acquired by NIS elements (Nikon) or Fluoview (Olympus), and were adjusted using Adobe Photoshop CS4 software (Adobe Systems, San Jose, CA). Each image resulting from immunofluorescence is representative of at least three independent experiments.

### siRNA-mediated silencing

On-Target plus SMARTpool siRNA against rat exophilin-8 (catalog no. 360034), RIM-BP2 (266780), RIM1 (84556), RIM2 (116839), myosin-Va (25017), myosin-VIIa (266714), and Ca$_v$1.3 (29716), as well as the control On-Target plus non-targeting pool siRNA, were purchased from GE Dharmacon. INS-1 832/13 cells were plated at a density of $2.5 \times 10^6$ in a 6-well culture plate and were grown for 24 hr. Suspended cells after trypsinization were transfected twice with siRNAs using Lipofectamine RNAiMAX reagent (Invitrogen), according to the manufacturer's instructions. The second transfection was performed 72 hr later, and the cells were analyzed 36 hr thereafter.

### Sucrose density gradient centrifugation

Sucrose density gradient centrifugation analysis was performed as described previously (*Fujita et al., 2009*). Briefly, INS1 832/13 cells were homogenized in homogenization buffer (50 mM Tris-HCl pH 7.5, 150 mM NaCl) containing complete protease inhibitor cocktail (Roche, Basel, Switzerland) by repetitively passing through 1-ml syringe with a 27-gauge needle. The homogenate was centrifuged at $10,000 \times g$ for 10 min, and the supernatant was centrifuged at $100,000 \times g$ for 60 min. Resulting 600 µl of the supernatant (cytosol fraction) was loaded on the top of discontinuous sucrose gradients. The gradient was composed of 1 ml each of stepwise concentration of sucrose solutions (3, 6, 9, 12, 15, 18, 21, 24, 27, 30% sucrose in homogenization buffer). The samples were centrifuged at $100,000 \times g$ for 18 hr, and then 600 µl fractions were collected from the top of the gradient.

### Statistical analysis

No samples or animals were excluded from the analysis. No statistical methods were used to predetermine sample size and experiments were not randomized. Statistical significance was determined using a two-tailed unpaired *t*-test. The investigators were not blinded to allocation during experiments and outcome assessment.

## Acknowledgements

We are grateful to T Nara and T Ushigome for their colony management of mice, and S Shigoka for her assistance in preparing the manuscript. This work was supported by JSPS KAKENHI Grant Numbers JP20113005, JP24390068, JP26670133, and JP2604104, and JP16K15211 to TI and JP15K08295 to KM. It was also supported by grants from Novo Nordisk Insulin Study Award (to TI) and from Banyu Life Science Foundation International (to KM).

## Additional information

### Funding

| Funder | Grant reference number | Author |
| --- | --- | --- |
| Banyu Life Science Foundation International | | Kohichi Matsunaga |
| Japan Society for the Promotion of Science | JP15K08295 | Kohichi Matsunaga |
| Japan Society for the Promotion of Science | JP20113005 | Tetsuro Izumi |
| Japan Society for the Promotion of Science | JP24390068 | Tetsuro Izumi |
| Japan Society for the Promotion of Science | JP26670133 | Tetsuro Izumi |

| Japan Society for the Promotion of Science | JP2604104 | Tetsuro Izumi |
| Japan Society for the Promotion of Science | JP16K15211 | Tetsuro Izumi |

The funders had no role in study design, data collection and interpretation, or the decision to submit the work for publication.

## Author contributions

FF, Data curation, Formal analysis, Investigation, Methodology; KM, Data curation, Formal analysis, Supervision, Funding acquisition, Validation, Investigation, Visualization, Methodology, Project administration; HW, Supervision, Investigation, Methodology; RI, Resources, Supervision, Investigation, Methodology; EK, Resources, Formal analysis, Investigation; HK, YM, Resources; KO, Resources, Supervision; TI, Conceptualization, Resources, Data curation, Supervision, Funding acquisition, Writing—original draft, Project administration, Writing—review and editing

## Author ORCIDs

Tetsuro Izumi, http://orcid.org/0000-0002-0974-7384

## Ethics

Animal experimentation: The protocol for animal experimentation was approved by the Institutional Animal Care and Use Committee of RIKEN Kobe Branch. All the animal experiments were conducted in accordance with the RIKEN institutional guideline and the rules and regulations of the Animal Care and Experimentation Committee, Gunma University (the approved number: 14-018).

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
