## [Decision Letter]

Thank you for submitting your article "Exophilin-8 assembles secretory granules for exocytosis in the actin cortex via interaction with RIM-BP2 and myosin-VIIa" for consideration by *eLife*. Your article has been favorably evaluated by Anna Akhmanova (Senior Editor) and three reviewers, one of whom, Suzanne Pfeffer, is a member of our Board of Reviewing Editors. One of the other two reviewers, John Hammer III, has agreed to share his identity..

The reviewers have discussed the reviews with one another and the Reviewing Editor has drafted this decision to help you prepare a revised submission.

Summary:

This is a well written study of vesicle docking in insulin secreting cells, including creating a knock out mouse missing the protein, Exophilin-8, and carrying out related phenotypic analysis that suggest that the Rab3A-binding, synaptic vesicle docking factor, Rim, functions with RIM-BP2, Myosin-VIIa, and actin, to concentrate vesicles at islet cell release sites. The work appears to be carried out to a high standard and will be of broad interest. Although the identification of interaction domains and localizations are standard cell biology, the inclusion of a mouse model and measurement of physiological consequences add in vivo significance to the story. Thus, presentation in *eLife* is recommended if the authors are able to address the following points.

Essential revisions:

1) The authors conclude that the complex is formed by direct interactions (exophilin-8 with RIMBP2 and RIMBP2 with myosinVIIa) but the data do not meet common standards. The data provided are co-immunoprecipitations followed by immunoblotting and with one or both partner transfected. In most cases the signals are strong and appear specific. Also, the involvement of SH3 and polyproline is highly suggestive of direct binding for one of the interactions. Nevertheless, to demonstrate direct interactions requires tests with purified proteins. The novel discovery here is RimBP2 functioning as an adaptor linking exophilin-8 to myosin VIIa – this needs to be shown with the corresponding purified fragments to meet the standard of *eLife*.

2) By the end of this study the co-IP data provide evidence for an incredibly large network of exophilin8 interactions: Rab27, RimBP2, myosinVIIa, PKA, Sec6, actin, Cav1.3, RIM1/2, synataxin1 and Munc13-1. Are these many separate complexes or one large complex? Are they present for endogenous rather than over-expressed exophilin-8? Is the exophilin8-RIMBP2-myosinVIIa complex present in the cytoplasm as a stable ternary subcomplex? The major finding of this study is the identification of the exophilin8-RIMBP2-myosinVIIa complex. Its presence as an endogenous complex should be defined and characterized biochemically (e.g. using gradient or column fractionation) – this may be discernible with a western blot across gradient or column fractions.

3) The RimBP2 knockdown phenotype is a critical aspect of the present study. Please show a blot for the RimBP2 knockdown. More importantly, other than a non-targeting siRNA pool, there is no specificity control. The central conclusion would be considerably strengthened with rescue of the knockdown phenotype by wildtype but not mutated versions that lack either binding to exophilin-8 or binding to myosin-VIIa.

---

## [Author Response]

*1) The authors conclude that the complex is formed by direct interactions (exophilin-8 with RIMBP2 and RIMBP2 with myosinVIIa) but the data do not meet common standards. The data provided are co-immunoprecipitations followed by immunoblotting and with one or both partner transfected. In most cases the signals are strong and appear specific. Also, the involvement of SH3 and polyproline is highly suggestive of direct binding for one of the interactions. Nevertheless, to demonstrate direct interactions requires tests with purified proteins. The novel discovery here is RimBP2 functioning as an adaptor linking exophilin-8 to myosin VIIa – this needs to be shown with the corresponding purified fragments to meet the standard of* eLife.

Because bacterially expressed recombinant proteins, particularly those with high molecular weight, are prone to be inappropriately folded, we employed a system where multiple proteins are simultaneously expressed in HEK293A cells, as previously described (Morita E., Cell Host Microbe, 2007; Matsunaga et al., J. Cell Sci., 2017). We first tagged wild-type or PA mutant exophilin-8 ABD with One-STrEP-Flag (OSF) and the binding proteins were pulled down using Strep-Tactin beads followed by SDS-PAGE and CBB staining (new Figure 9). As you see, the exophilin-8 ABD, but not its PA mutant, binds RIM-BP2, and there are no other specific major bands (compare lanes 2 and 5), which strongly suggests the direct binding between exophilin-8 and RIM-BP2 (some other proteins do not mediate the complex formation). We also found that, although the PA mutant does not interact with myosin-VIIa in INS1 832/13 cells (Figure 8), both the WT and the PA mutant bind myosin-VIIa even without RIM-BP2 expression in HEK293A cells (lanes 3, 4, 6, and 7). This finding indicates that exophilin-8 seems to directly interact with myosin-VIIa at least in heterologous cells, which is consistent with the previous findings (El-Amraoui et al., 2002; Fukuda and Kuroda, 2002). Furthermore, exophilin-8 binds myosin-VIIa without its whole MBD or the proline-rich motif in its ABD. We next expressed OSF-RIM-BP2 and the binding proteins were examined (new Figure 9). As expected, RIM-BP2 bound WT exophilin-8 ABD, but not to the PA mutant (lanes 2 and 3). However, RIM-BP2 could not bind myosin-VIIa without simultaneous expression of the wild-type exophilin8 ABD (lanes 4-6), indicating that RIM-BP2 cannot directly bind myosin-VIIa. Please note that we have not claimed the direct interaction between RIM-BP2 and myosin-VIIa, in contrast to that between exophilin-8 and RIM-BP2, in the original manuscript. We, however, noticed that, although the approximate molecular mass by SDS-PAGE of myosin-VIIa expressed in HEK293A cells is ~260 kDa corresponding to its calculated molecular mass (Figure 9), that of myosin VIIa interacting with RIM-BP2 in β-cell lines is ~190 kDa (Figure 8). This 190-kDa protein should be myosin-VIIa, because it was identified with RIM-BP2 from the 150~190 kDA band of exophilin-8-interacting proteins in MIN6 cells by LC-MS/MS (Figure 3), and more importantly, because both 260-kDa and 190-kDa proteins were downregulated by siRNA specific to myosin-VIIa (new Figure 9) and reacted with two different kinds of anti-myosin-VIIa antibodies (Figure 6, Figure 8, Figure 9 and new Figure 9). Therefore, the 190-kDa form of myosin-VIIa appears to be specifically expressed in β-cells. When we performed the pull-down assays in INS1 832/13 cells, we found that only the 190-kDa protein was pulled down with RIM-BP2, although the 260-kDa protein was also pulled down after exogenous expression of myosin-VIIa (Figure 9). These findings indicate that the 260-kDa protein is derived from transfected myosin-VIIa cDNA and interacts with RIM-BP2 through exophilin-8, because it can directly interact with exophilin-8 but not with RIM-BP2 (Figure 9). Because the amount of 190-kDa myosin-VIIa pulled down with OSF-RIM-BP2 was not markedly increased compared with that of 260-kDa myosin-VIIa, after exogenous expression of myosin-VIIa, it may represent an alternatively spliced form that specifically functions in secretory cells. Although the exact molecular nature of 190-kDa myosin-VIIa and the functional significance of the direct interaction of exophilin-8 and 260-kDa myosin-VIIa in β-cells are not uncovered in this study, we showed that either the PA mutant that disrupts the interaction of exophilin-8 with 190-kDa myosin-VIIa through RIM-BP2, or the ΔSH3 mutant (new Figure 6) that disrupts the interaction of RIM-BP2 with exophilin-8 and myosin-VIIa and should not affect the direct interaction between exophilin-8 and 260-kDa–myosin-VIIa, does not restore the peripheral granule accumulation or rescue the decreased insulin secretion in knockdown or knockout cells (Figure 6, and see below in the essential revision 3 and new Figure 7) strongly indicates that the exophilin-8–RIM-BP2–190-kDa myosin-VIIa complex formation is functionally critical for granule exocytosis.

*2) By the end of this study the co-IP data provide evidence for an incredibly large network of exophilin8 interactions: Rab27, RimBP2, myosinVIIa, PKA, Sec6, actin, Cav1.3, RIM1/2, synataxin1 and Munc13-1. Are these many separate complexes or one large complex? Are they present for endogenous rather than over-expressed exophilin-8? Is the exophilin8-RIMBP2-myosinVIIa complex present in the cytoplasm as a stable ternary subcomplex? The major finding of this study is the identification of the exophilin8-RIMBP2-myosinVIIa complex. Its presence as an endogenous complex should be defined and characterized biochemically (e.g. using gradient or column fractionation) – this may be discernible with a western blot across gradient or column fractions.*

We have investigated the complex formation in INS1 832/13 cells using sucrose density gradient centrifugation (new Figure 9—figure supplement 1). We have focused on exophilin-8, RIM-BP2, and myosin-VIIa, because only a portion of exophilin-8 is thought to be linked to the Ca^2+^ channel and the exocytic machinery along the plasma membrane as we originally discussed. Endogenous exophilin-8 (130 kDa) and RIM-BP2 (150 kDa) exhibited similarly wide distributions in fractions 3-10, whereas both endogenous 190-kDa and 260-kDa myosin-VIIa showed relatively narrower distributions around fractions 6-10. Judged from the positions of molecular mass markers, these findings are consistent with the ternary complex formation among these proteins in the cells, although each protein also appeared to exist as a monomer. Because multiple forms of complexes can exist among exophilin-8, RIM-BP2, 190 kDa myosin-VIIa, and 260 kDa myosin-VIIa with or without additional proteins, such as Ca_v_1.3 and RIM in cells, it is difficult to get further information by this biochemical method.

*3) The RimBP2 knockdown phenotype is a critical aspect of the present study. Please show a blot for the RimBP2 knockdown. More importantly, other than a non-targeting siRNA pool, there is no specificity control. The central conclusion would be considerably strengthened with rescue of the knockdown phenotype by wildtype but not mutated versions that lack either binding to exophilin-8 or binding to myosin-VIIa.*

We originally showed a blot for RIM-BP2 knockdown in Figure 6-supplement 1E, but have shown in Figure 7 and described it clearly on page 11 in the revised manuscript. For rescue experiments, we first mutated mouse wild-type RIM-BP2 cDNA so that it will become resistant to one of the rat RIM-BP2 siRNA whose target sequence is conserved in mouse RIM-BP2. We then generated ΔSH3 mutant that lacks all the SH3 domains, and found that this mutant loses the binding activity to both exophilin-8 and myosin-VIIa (Figure 6). We made this deletion mutant, because there is no known amino acid mutation within the SH3 domain that commonly disrupts interaction with its specific protein ligand, although it binds short proline-rich peptide segments within proteins. Protein ligands of SH3 domain exploit multiple discontiguous interactions to enhance affinity and selectivity, and the amino acid similarity of any two SH3 domains is typically 25% (see Reviews, Dolgarno et al., Biopoly, 1997, Kay, FEBS Letters, 2012). When expressed in RIM-BP2 knockdown cells, only wild-type RIM-BP2, but not the ΔSH3 mutant, rescued the decreased insulin secretion and restored the peripheral accumulation of insulin granules (new Figure 7, Figure 7—figure supplement 2, Figure 7—figure supplement 3).